



# Identification of atmospheric and oceanic teleconnection patterns in a 20-year global data set of the atmospheric water vapor column measured from satellites in the red spectral range

Thomas Wagner[1] and Steffen Beirle[1], Steffen Dörner[1], Christian Borger[1], Roeland Van Malderen[2]

[1]Satellite Remote Sensing Group, Max Planck Institute for Chemistry, Mainz, Germany
[2]KMI – IRM, Royal Meteorological Institute of Belgium, Brussels, Belgium

*Correspondence to*: Thomas Wagner (thomas.wagner@mpic.de)

**Abstract.** We used a global long-term (1995-2015) data set of total column water vapor (TCVW) derived from satellite observations to quantify the influence of teleconnections. To our knowledge, such a comprehensive global TCWV data set was rarely used for teleconnection studies. One important property of the TCWV data set is that it is purely based on observational data. We developed a new empirical method to decide whether a teleconnection index is significantly detected in the global data set. Based on this method more than 40 teleconnection indices were significantly detected in the global

TCWV data set derived from satellite observations. In addition to the satellite data we also investigated the influence of teleconnection indices on other global data sets derived from ECMWF reanalysis (ERA). One important finding is that the results obtained for the ERA TCWV data are very similar to the observational TCWV data set indicating a high consistency between the satellite and ERA data. Moreover, similar results are also found for two selections of ERA data (either all data or mainly clear sky data). This finding indicates that the clear-sky bias of the satellite data set is negligible for the results of

this study. For most 'traditional' teleconnection data sets (surface temperature, surface pressure, geopotential heights and meridional winds at different altitudes) a smaller number of significant teleconnection indices was found than for the TCWV data sets, while for zonal winds at different altitudes, the number of significant teleconnection indices (up to >50) was higher. In all global data sets, no 'other indices' (solar variability, stratospheric AOD or hurricane frequency) were significantly detected. Since many teleconnection indices are strongly correlated, we also applied our method to a set of

orthogonalised indices. The number of significantly detected orthogonalised indices (20) was found to be much smaller than for the original indices (42). Based on the orthogonalised indices we derived the global distribution of the cumulative influence of teleconnection indices. The strongest influence on the TCWV is found in the tropics and high latitudes.

## 1 Introduction

It has been known for a long time that weather at one location can be linked to weather at a far distant location (Walker and Bliss, 1932; Bjerknes, 1966, 1969; Wallace and Gutzler, 1981; Nigam and Baxter, 2015; Feldstein and Frantzke, 2017 and references therein). The distances between such locations can be very large, up to opposite locations on the globe. The strength of the correlation varies with location exhibiting regions of maximum (anti-) correlations and regions without any

significant correlation. The resulting correlation patterns are referred to as teleconnection patterns. The strongest teleconnection is the El-Nino / Southern oscillation (ENSO) phenomenon (Walker and Bliss, 1932; Bjerknes, 1966, 1969), but many more teleconnection are known, which are located in many regions on both hemispheres (e.g. Feldstein and Frantzke, 2017 and references therein).

The temporal variability of teleconnections is usually described by teleconnection indices (e.g. the ratio of surface pressures

at selected stations) and covers a wide range of frequencies from a few days to inter-annual and inter-decadal time scales (Hurrel, 1995; Feldstein, 2000; Nigam and Baxter, 2015; Woolings et al., 2015; Feldstein et al., 2017). Atmospheric teleconnections (like e.g. the North Atlantic Oscillation, NAO) have typically higher intrinsic frequencies than oceanic teleconnection indices (like e.g. the Atlantic Meridional Mode, AMM).



Teleconnections can be identified in different data sets like sea level pressure, surface air temperature, sea level pressure as well as geopotential heights and wind fields at different altitudes (Wallace and Gutzler, 1981; Thompson and Wallace, 1998; Nigam and Baxter, 2015; Feldstein and Frantzke, 2017). In recent studies, the geopotential height is the most used quantity for the quantification of teleconnections. Teleconnections are mainly found in the troposphere with the strongest amplitudes in the upper troposphere (Feldstein, 2000). But several teleconnections have also connections to the stratosphere (Feldstein, 2000 and references therein; Nigam and Baxter, 2015; Feldstein and Frantzke, 2017; Domeisen et al., 2019).

Teleconnections can be identified and defined in different ways: historically, teleconnection indices were empirically and intuitively determined based e.g. on the locations of meteorological stations (e.g. Walker and Bliss, 1932). In later studies more objective methods were developed based on correlation matrices, principle component analyses (PCA) (also referred to as empirical orthogonal function (EOF) methods) or rotated PCA (also referred to as varimax rotation). More details about these and further methods can be found in Horel (1981), Wallace and Gutzler (1981), Barnston and Livezey (1987),

Thompson and Wallace (1998), Feldstein and Frantzke (2017) and references therein. If these methods are applied, the derived teleconnections time series and spatial patterns particularly depend on the selected region of the globe (e.g. northern hemisphere) and the selected season (e.g. winter months). Usually, these methods are not applied for the full globe.

Besides the fact that teleconnections are interesting in themselves, their study is also important for other applications. For example, taking teleconnections into account can improve weather forecasts (Feldstein and Frantzke, 2017 and references

therein). They have impact on extreme events, e.g. heat waves, droughts, and floods (King et al., 2016; Yeh et al., 2018 and references therein) and can affect storm tracks. In addition to atmospheric quantities (e.g. humidity, precipitation, stratospheric ozone), teleconnections also affect oceanic variables (e.g. Arctic and Antarctic sea ice, the Atlantic thermohaline circulation) and the marine and terrestrial ecosystems (Feldstein and Frantzke, 2017 and references therein). Finally it is worth noting that teleconnections are expected to change in a changing climate (e.g. King et al., 2016; Feldstein

and Frantzke, 2017; Yeh et al., 2018).

In this study we investigate the influence of various teleconnections on the global distribution of the total column water vapor (TCWV). For that purpose we use a consistent long term data set (1995 – 2015) derived from satellite observations in the red spectral range obtained from GOME on ERS-2, SCIAMACHY on ENVISAT and GOME-2 on METOP (Beirle et al., 2018). The data sets consists of monthly mean values on a 1° x 1° latitude/longitude grid, which were carefully merged

making use of the long overlap time between the different satellite data sets (for details see Beirle et al., 2018). Validation by independent data sets showed a smooth temporal variation with a stability within 1% over the whole period (1995-2016) (Danielczok and Schröder, 2017). To our knowledge, teleconnection studies using water vapor data sets are rare (e.g. van Malderen et al., 2018). One particular speciality / advantage of our study is that we use for the first time a global data which is entirely based on measurements. Here it is important to note that the TCWV is dominated by the atmospheric layers close

to the surface. Another important aspect of our study is the development of a new empirical method to decide whether a teleconnection (index) can be significantly identified in an atmospheric data set or not.

Our study addresses the following main questions:

a) Which teleconnection index (and other time series like indices of solar activity) can be significantly identified in the satellite TCWV data set (or other data sets)?

b) Are the same results obtained for TCWV data from observations and models? Here also the question is addressed how representative the satellite observations (for mainly clear sky) are for all sky data sets.

c) How does the number of significant teleconnections in the global TCWV data sets compare to similar results obtained for "traditional" teleconnection data sets like surface temperature, sea level pressure or wind fields and geopotential heights at different altitudes?

d) What is the spatial distribution of the influence of teleconnection patterns on the global TCWV distribution?



The paper is organised as follows: In section 2 the global data sets used in this study, and in section 3 the considered (mostly teleconnection) indices are introduced. Section 4 presents the fit function of the indices to the global data sets and the obtained global patterns. In section 5 a new method for the determination of the significance is introduced, which is applied to the different global data sets in section 6. In section 7 a reduced set of orthogonalised teleconnection indices is extracted

and. Section 8 presents the global distribution of the cumulative influence of the teleconnection indices.

**2 Data sets**

**2.1 Total water vapor column**


Our study focuses on global long term data sets of the total column water vapor (TCWV). Here we use three data sets:

a) Satellite observations from July 1995 to October 2015 (Beirle et al., 2018) derived from the satellite instruments GOME on ERS-2 (1995 to 2003), SCIAMACHY on ENVISAT (2002 to 2012) and GOME-2 on MetOp (2006 to present), which have similar overpass times (between 9:30 and 10:30 LT). The start date of the time series was predetermined by the start of

the first satellite mission; the end date of the time series was set to October 2015, because some of the used time series were only available until that date. The data set is available on a 1° x 1° latitude/longitude grid with monthly resolution. The data set does not cover polar winter, since the satellite observations use scattered and reflected sun light.

In Fig. 1 the variation of the TCWV with latitude and time is shown (the latitude bins represent zonally averaged values). The top panel shows the original TCWV data set, whereas both lower panels present the absolute and relative anomalies with

the mean seasonal cycle removed. Several anomaly patterns are clearly obvious, which are mainly related to strong ENSO events. Especially for the relative anomalies, many high frequency variations are found. While part of these high frequency variations represent measurement noise and atmospheric noise, the results of this study showed that they also represent atmospheric teleconnections.

In addition to the satellite observations of the TCWV we also use global time series of the TCWV derived from ECMWF

reanalysis (ERA Interim, Dee et al., 2011). Here we use two data sets:

a) All ERA data including clear and cloudy conditions

b) Only ERA data for clear sky observations. Here, a cloud cover below 0.3 between 1km and 6km is regarded as cloud free. This criterion reflects the observational conditions of the satellite data set.

For both ERA data sets, the TCWV was temporally interpolated to the time of the satellite overpass (10:00 LT). From the

comparison of the results for the measurements and model data sets, the effect of the specific sampling of the satellite observations (which represent only clear sky observations) can be investigated. In Fig. 2 the global mean distributions of the TCWV data sets from satellite observations and ERA data are shown.

**2.2 Other global data sets**


Teleconnections patterns are usually derived from meteorological quantities like surface pressure and temperature or geopotential heights and wind fields at different altitudes. In this study we also consider such quantities, which we also obtained from ERA data (see Table 1). We analyse these data sets similarly to the TCWV data sets (details are described below). In this way we will assess in how far the impact of teleconnections on TCWV is comparable to traditional

teleconnection data sets. In Figs. A1 and A2 in the appendix, the global mean distributions of all data sets are shown.



**3 Teleconnection indices**

We performed an extensive search for teleconnection indices in the scientific literature and web sites of national weather services. We found in total 54 teleconnection indices, which cover the time span of our TCWV data set. An overview on these teleconnection indices as well as additional time series (e.g. of the solar activity) is given in Table 2. Although we not only focus on teleconnection indices in this study, in the following we use the term 'index' to describe the whole set of teleconnection indices and other time series.

It should be noted that for several teleconnection indices (in particular for the Madden-Julian oscillation) different definitions exist. Thus the number of teleconnection indices in Table 2 is much larger than the corresponding atmospheric phenomena. A more detailed overview on the selected indices and their data sources is provided in Fig. A3 in the appendix.

Before the indices are fitted to the different global data sets, the mean seasonal cycle (1995 – 2015) is subtracted (like for the data sets themselves, see Fig. 1). Some teleconnection indices are characterised by a strong seasonal cycle, whereas others

are not. In addition, also a linear trend is fitted and subtracted. Finally the obtained anomalies are normalised by the corresponding standard deviations. This ensures that the obtained fit coefficients for the different indices can be directly compared. The different steps of these preparations are illustrated in Fig. 3. It is interesting to note that many of the considered teleconnection indices are highly correlated. Fig. 4 presents a matrix with correlation coefficients between the different indices (after the seasonal cycles were removed).


**4 Analysis of global data sets**

To determine the strength with which individual indices are detected in the temporal variations of the different global data sets, they are fitted to the global data sets.


**4.1 Fit function**

For each 1° x 1° latitude / longitude pixel of the global data sets the time series of the monthly mean anomalies of the global data sets (the example below is for the TCWV) are fitted by the following function:

$$TCWV(t) = c + b \cdot t + f_i \cdot index_i(t) \tag{1}$$

Here c and b describe constant and linear terms. $index_i$ represents the selected normalised index of monthly mean anomalies. The fit coefficient $f_i$ describes the sign and strength of the contribution of the chosen index to the variability of the TCWV anomaly of the chosen 1° x 1° pixel. The fit function is separately applied to the individual indices listed in Table 2. Here it

should be noted that the fit function could in principle be applied to several or even all indices simultaneously. However, since many indices are highly correlated, the interpretation of the results would then not be straight forward. Thus, we chose to include the individual indices one by one in the fit function. Besides the parameters c, b, and the fit coefficient $f_i$, also the difference between the temporal variation of the global data sets and the fit function is quantified by the root mean square (RMS). In addition to the fit function described in equation 1, a second fit is performed with only the constant and linear

terms:

$$TCWV(t) = c + b \cdot t \tag{2}$$

The comparison of the RMS with and without including the index term allows to quantify the importance of the chosen

index to describe the temporal variation of the data set. This RMS differences is then divided by the zonal mean value of the considered quantity, because (like for water vapor) many of the analysed quantities depend strongly on latitude (see equation 3). In the following this quantity is referred to as delta RMS.





$$delta\ RMS = \frac{RMS_{without\ index} - RMS_{with\ index}}{mean\ of\ data\ set(latitude)} \quad\quad (3)$$

In Fig. 5 fit results for the ENSO index are shown derived for the TCWV from satellite observations (left), ERA data (center), and ERA data for clear sky conditions (right). The results for the other data sets will be discussed in section 6. The top panel in Fig. 5 shows the fit coefficients for the ENSO index. High fit coefficients mean that part of the measured TCWV time series can be explained by the ENSO index pattern. High negative fit coefficients mean the same for the negative ENSO index. Fit coefficients of zero indicate no connection to ENSO. Very similar spatial patterns are found for the three TCWV

data sets indicating that the ENSO phenomenon is well captured in the satellite and model data sets. From the similarity between the model data including all sky conditions (center) or only clear sky conditions (right) it can be concluded that the satellite observations (representing mainly clear sky conditions) are representative for all sky conditions (no obvious clear sky bias).

   The second row in Fig. 5 presents the normalised RMS of the differences between the measurements and the fit functions.

Note that in order to account for the strong latitudinal dependence of the TCWV, the RMS are normalised for each latitude bin by the mean values for all longitudes of the considered data sets. In all three data sets, the smallest RMS are found close to the equator. This is an interesting finding, but can probably be explained by a) the rather high TCWV and b) its rather small variability in these regions. In mid-latitudes, systematically higher RMS are found for the satellite observations compared to the model results. This is probably related to the rather large effects of clouds on the satellite observations,

which becomes especially important in these regions (clouds lead to less valid observations and larger measurement uncertainties). Another interesting finding is that in polar regions the RMS for the satellite observations is smaller than for the model results. This finding is probably related to the sparseness of water vapor measurements in these regions assimilated in the ECMWF model. Thus the spatio-temporal variability of the satellite observations is probably more realistic than that of the model data. The RMS for the model results for clear sky conditions is slightly higher than for the

model results for all conditions, which is to be expected because of the reduced number of data available for the cloud-filtered data set.

   The lower panel of Fig. 5 shows the delta RMS for the ENSO index indicating the reduction of the RMS if the ENSO index is included in the fit. As expected, the largest delta RMS is found over the tropical Pacific, where the ENSO phenomenon is most pronounced. The global distribution of the delta RMS is very similar for the three data sets. The fit coefficients and

delta RMS for three other selected indices are shown in Fig. 6 for the TCWV data set from satellite observations. For all indices, specific activity centers can be found in different parts of the globe. The fit coefficients and delta RMS for all indices are presented in the appendix (Fig. A4). Note that very similar spatial patterns are found for the three TCWV data sets.

   It should be noted that in many teleconnection studies (e.g. Horel, 1981), the strength of a teleconnection index is quantified

by calculating the ratio of the difference of the RMS (with and without an index included) and the total RMS. In this study we applied a different procedure as described above, because the total RMS depends on many factors, in particular also on the uncertainties of the considered data set. Since we want to compare the delta RMS values derived for different data sets (in particular the TCWV data sets derived from satellite observations and model results, but also other datasets) in a quantitative way, we decided to divide the RMS (with and without an index included) by the zonal mean of the considered

data set. Thus the delta RMS shows the relative impact of the respective index. While the RMS of the different TCWV data sets are rather different (see Fig. 5, middle panel), the zonal means are very similar (Fig. 2). The zonal mean was chosen (instead of the long term average of each considered 1° x 1° pixel), because for some data sets used in this study (especially the wind data sets) large variations and even zero-crossings exist, which would lead to meaningless delta-RMS values. We compared the delta RMS values calculated by our new definition with those of the more traditional definition for the TCWV

data sets (Fig. A5 in the appendix). The obtained global patterns of both delta RMS definitions are almost identical.





**4.2 Quantification of the strength of a (teleconnection) index**

For a quantitative assessment of the strength and significance of a (teleconnection) index, we calculated the 99th percentile

of fit indices $f_i$ for all 1° x 1° pixel values. We chose the 99[th] percentile because it is close to the maximum, but still not affected by individual outliers. Fig. 7 presents the 99th percentiles (p99) for all considered indices for the three TCWV data sets. For all three TCWV data sets the highest p99 values are found for the ENSO-like teleconnection indices.

**5 Determination of significance**


For most teleconnection indices clear spatial patterns of fit coefficients and delta RMS values are found in the global maps (see Fig. A4) indicating that these indices are significantly detected in the global water vapor data sets. However, from the fit results themselves it is not easily possible to judge about the significance of the detection of an index, mainly because the effects of atmospheric noise and other uncertainties of the data sets cannot easily be quantified (see also Wallace and

Gutzler, 1981).

To address these difficulties, we developed an empirical approach to determine threshold values for the p99 values. If the p99 values for a given teleconnection index is above the threshold, the index is considered as significantly detected in the considered data set. It is clear that also with this approach, for indices with p99 values close to the threshold value no clear decision about the significance can be made. The advantage of the new approach is, however, that it provides a clear

procedure and in particular a metric which allows a quantitative comparison between different data sets (see section 6).

**5.1 Use of reversed indices**

Our approach for the estimation of the significance level is based on the use of reversed indices. The basic idea is that the

reversed indices should not contribute to the temporal variation in the global data sets, because they have no geophysical basis. Thus the derived delta RMS and p99 values can be used as an estimate of the detection limit for the significance of a fitted index for the given measurement errors of input data. If the p99 values are above the threshold, it is likely that the considered index significantly contributes to the variability of the considered data set. For the determination of the detection limit we take into account the reversed indices of all original indices used in this study (see Table 2 and Fig. A3). This

approach has the advantage that all relevant frequencies of real indices and teleconnection indices are considered.

In Fig. 7 besides the p99 values for the original indices (blue), also those for the reversed indices are shown (red). The p99 values for the reversed indices are much smaller than most of the original indices. Interestingly, the variability of the p99 values for the reversed indices is rather high (Fig. 8). In Appendix 1 the reasons for this variability will be further investigated.


**5.2 Effect of shifts of the (teleconnection) indices**

In addition to the p99 values themselves, also the effect of time shifts Δt = ± 1 month of the indices on the p99 values was considered to decide whether an index was significantly identified in a global data set, because for indices with a geophysical

relationship to a considered data set, the exact temporal synchronisation should be important (but might depend on region). In contrast, for indices without a geophysical relationship to the considered data set, the p99 values should not depend on the exact temporal synchronisation. In Fig. A8 the p99 values for the original and shifted (by ± 1 month) indices are shown for the TCWV data set from satellite observations. For most data sets (especially for those with high p99 values) indeed smaller





p99 values are found for the shifted indices. Here it is interesting to note that in general a stronger effect is found for atmospheric indices than for oceanic indices, which can be understood by the higher frequencies of the atmospheric indices. For several oceanic indices, even higher values are found for the shifted indices indicating a time shift (mostly a time lag) between the TCWV and these indices. For one index (AMM) higher p99 values are even found for shifts in both directions indicating an ambiguity in the synchronisation between the TCWV and the AMM index.

Another interesting finding is that for some atmospheric indices with p99 values below the significance threshold (PE, MJ2,
OOMI2, FMO1) still rather small ratios of the shifted and original indices are found indicating that these indices are also probably significantly detected in the TCWV data set. Thus in the following we consider also indices with p99 values below the significance threshold but with p99 ratios below 0.8 for both shifts as significantly detected. Here it should be noted that the choice of the threshold value of 0.8 is somehow arbitrary. It was chosen because a deviation of 20% from unity is larger than the 'noise level' of the ratio. The exact choice of the threshold has only a small effect on the obtained results. For the
TCWV data set from satellite observations, this additional criterion increases the number of significantly detected indices from 40 to 42. For the ERA TCWV data sets the number of significantly detected indices increases from 43 to 44 (and from 39 to 42 for ERA data for clear sky conditions).

## 6 Comparison of the number ‚significant indices' for the different global data sets


A rather high number of significant indices was identified in the global TCWV data sets. To put this finding into a broader perspective, we applied the same procedure also to other global data sets, which are usually considered in teleconnection studies (see Table 1). The corresponding p99 values of the different indices (including also the reversed indices) are presented in Fig. A9. In general similar results as for the TCWV data sets are found. In particular, for all data sets a large
number of teleconnection indices is significantly detected. However, also differences are found: in particular, the teleconnection index with the maximum p99 value is found to be different for the different data sets. A summary of the number of significant indices and the teleconnection index with the highest p99 is given in Table 3. Most significant indices are found for the zonal winds with the highest number in the upper troposphere. For these data sets the number of significant indices is larger than for the TCWV data sets. For geopotential heights and meridional winds, less significant indices are
found (and even less than for the TCWV data sets). For geopotential heights most significant indices are found in the upper troposphere, while for the meridional winds no clear altitude dependence is observed. Also for the surface temperature and surface pressure rather low numbers (less than for the TCWV data sets) of significant indices are found. From these results we conclude that the global TCWV data sets are well suited for teleconnection studies.

It is also interesting to note that for different groups of data sets specific indices with maximum p99 values were found:
ENSO-like indices were found for the water vapor data sets, surface temperature and for zonal winds at most altitudes (except for 950 hPa, for which AO has the highest p99 value); AAO was found for surface pressure, geopotential heights and meridional winds at 500, 200, and 50 hPa, and SCA for meridional winds at 850 and 950 hPa.

## 7 Derivatives and orthogonalised indices


It was shown in Fig. 4 that many indices are strongly correlated. Thus the numbers of ‚significant indices' obtained in the previous chapters are not useful to represent the number of independent significant indices. This effect can be addressed by orthogonalisation of the indices before they are used for the analysis of the global data sets. Since it was found in section 5 (see Fig. A8) that for some indices time shifts led to higher p99 values, we also added the temporal derivatives of index
patterns to the list of indices to be orthogonalised. The p99 values for the temporal derivatives are shown in Fig. 9 (top). In general they are much smaller than the p99 values for the corresponding original indices, but still many p99 values were





found to be above the significance threshold. For the orthogonalisation, all ‚significant' original indices and temporal derivatives were considered (in total 57 indices). The orthogonalisation order was from highest p99 values to lowest p99 values. In Fig. 9 (bottom) the p99 values for the orthogonalised indices are shown. Compared to the results for the original
indices, two findings are of special importance:

-the number of significant indices (20) is much smaller than for the original indices (40) confirming that many teleconnection indices are indeed highly correlated and related to the same phenomena.

-the difference between the highest p99 value (for the ONI index) and subsequent p99 values is much larger than for the original indices. This finding indicates that the temporal pattern of the ENSO phenomenon is contained in many
teleconnection indices.

The delta RMS maps for the significant orthogonalised indices (together with the delta RMS maps for corresponding original indices) are presented in Fig. A10. Only one temporal derivative (of ONI) was found to be significant.

**8 Global distributions**


The delta RMS maps derived for the individual indices show characteristic patterns which indicate in which regions of the globe the selected index is important or not (see Fig. A4). In order to assess the global distribution of the general importance of teleconnections, we added the delta RMS maps of all significant indices to the figure. Maps of the derived cumulative delta RMS distributions are presented in Fig. 10 for different selections of teleconnection indices and TCWV data sets. In the
upper panel the patterns of all significant teleconnection indices found for the TCWV data set from satellite observations are added. In the middle panel the same is shown for the significant orthogonalised indices. The comparison again clearly indicates that many indices are highly correlated to the ENSO index. Thus, if only the orthogonalised indices are considered, the ENSO pattern, especially in the tropical Pacific, becomes relatively weaker compared to the cumulative delta RMS values in other regions. In the lower panel the cumulative delta RMS map for all significant orthogonalised indices for the
ERA TCWV data set is shown. The derived spatial patterns are very similar to those for the satellite data set. It should, however, be noted that also for regions in high latitudes, which are not covered by the satellite observations high values are found.

Fig. 11 shows the latitudinal (top) and longitudinal (bottom) distribution of the p99 values for all significant original indices (red) and all significant orthogonalised indices (blue). As expected, the highest values (related to ENSO) are found over the
equatorial east Pacific, but most indices have the strongest effects in mid and high latitudes. Interestingly, in the latitude range between −30° and +30° only for one significant orthogonalised index (besides ENSO) the maximum delta RMS is found.

**9 Conclusions**


We investigated the influence of a large set of teleconnection indices on the spatio-temporal variability of a global data set of the total column water vapor (TCWV) from 1995 – 2015 derived from satellite observations. To our knowledge, it is the first time that a global TCWV data set was used in such a detailed way in teleconnection studies (note that part of this data set was already used by van Malderen et al., 2018). Here it is important to note that the TCWV data set is purely based on
observational data. Another important achievement of this study is the development of a new empirical method to decide whether a teleconnection index is significantly detected in the global data set. The method is based on temporally reversed teleconnection indices, which ensures that all relevant time scales are considered. The new method can be applied in a universal way to different data sets.



Based on this method more than 40 teleconnection indices were significantly detected in the global TCWV data set derived
from satellite observations. Very similar results were obtained for two TCWV data sets from the ECMWF reanalysis (one
data set uses all data, the other only clear sky data points). From these findings we conclude 1) that the spatio-temporal
variability is well captured by both the satellite observations and the ERA Interim data, and 2) that a possible clear sky bias
is negligible.

We also applied the method to other data sets derived from the ECMWF reanalysis and compared the results to those for the
TCWV data sets. For most 'traditional' teleconnection data sets (surface temperature, surface pressure, geopotential heights
and meridional winds at different altitudes) less teleconnection indices were significantly detected than for the TCWV data
sets, while for zonal winds at different altitudes, more significant teleconnection indices (up to >50) were significantly
detected. For most data sets the strongest teleconnection signals were found for ENSO or AAO. For some data sets the
strongest teleconnection signals were found for SCA and AO. It should be noted that in none of the global data sets, other
indices (like the solar variability, the stratospheric AOD or the hurricane frequency) were significantly detected.

Since many teleconnection indices are strongly correlated, we also applied our method to the orthogonalised indices.
Compared to the original indices, much less orthogonalised indices (20 compared to 42) were significantly detected in the
TCWV data set from satellite observations. We investigated the spatial patterns of these orthogonalised indices and found
the strongest influence on the TCWV in the tropics and high latitudes.


**Acknowledgements**

We want to thank the European Centre for Medium-Range Weather Forecasts (ECMWF) and many scientific and national
institutions for making meteorological data and teleconnection indices available. We also thank ESA and EUMETSAT for
making satellite spectra available.

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





**Tables**

**Table 1: Meteorological data sets used in this study**

| Quantity | Source | Altitude |
|---|---|---|
| Water vapor VCD | Satellite observations | Total colum |
| Water vapor VCD | ECMWF reanalysis (all sky conditions) | Total colum |
| Water vapor VCD | ECMWF reanalysis (clear sky conditions) | Total colum |
| Surface temperature | ECMWF reanalysis (all sky conditions) | Surface |
| Surface pressure | ECMWF reanalysis (all sky conditions) | Surface pressure extrapolated to sea level |
| Geopotential heights | ECMWF reanalysis (all sky conditions) | 50hPa, 200hPa, 500hPa, 850hPa, 950hPa |
| Zonal winds | ECMWF reanalysis (all sky conditions) | 50hPa, 200hPa, 500hPa, 850hPa, 950hPa |
| Meridional winds | ECMWF reanalysis (all sky conditions) | 50hPa, 200hPa, 500hPa, 850hPa, 950hPa |

*the zonal winds at 50hPa are not further analysed, because they are – by definition - dominated by the QBO teleconnection
signal.


**Table 2: Teleconnection indices and other time series used in this study.. More details about these indices as well as their sources are given in Fig. A3 in the appendix.**

| Oceanic indices (23) | | Atmospheric indices (31) | | Others indices (7) |
|---|---|---|---|---|
| BEST | CAR | SCA | RMM1 | Solar indices: |
| N34 | AMO | AAO | VPM1 | RI |
| TPI | DMI | EAWR | MJ1 | MGII |
| ONI | AMM | NAO | MJ2 | SWO |
| ENSO | STA | EA | OOMI2 | S107 |
| N4 | TSA | PNA | OOMI1 | AP |
| HAW | EA_ersst | EPNP | FMO1 | |
| PDO | | SOI | FMO2 | HUR |
| PMM | | AO | Q30 | (hurricane |
| IND | | PE | QBO | frequency) |
| N1 | | WP | VPMN | |
| TNI | | NOI | FMON | |
| NTA | | VPM2 | OOMIN | SAOD |
| TNA | | RMM2 | MJN | (stratospheric |
| WHWP | | Q70 | RMMN | AOD) |
| IPO | | Q50 | | |








**Table 3: Numbers of significant indices and most significant indices for all data sets (the number of indices with p99 values below threshold but shift ratios <0.8 are indicated in brackets). The complete list of significant indices for the different data sets is provided in Table A1 in the appendix.**

| Data set | Number of significant indices | Most significant index |
|---|---|---|
| TCWV sat | 42 (2) | ONI |
| TCWV ERA | 44 (1) | ONI |
| TCWV ERA clear | 42 (3) | ONI |
| Tsurf | 37 (1) | ONI |
| Spred | 35 (1) | AAO |
| Geopot 50 hPa | 17 (5) | AAO |
| Geopot 200 hPa | 40 (0) | AAO |
| Geopot 500 hPa | 32 (1) | AAO |
| Geopot 850 hPa | 33 (1) | AAO |
| Geopot 950 hPa | 30 (1) | AAO |
| Zonal winds 200 hPa | 51 (0) | BEST |
| Zonal winds 500 hPa | 49 (0) | BEST |
| Zonal winds 850 hPa | 46 (1) | BEST |
| Zonal winds 950 hPa | 42 (4) | AO |
| Meridional winds 50 hPa | 24 (3) | AAO |
| Meridional winds 200 hPa | 32 (1) | AAO |
| Meridional winds 500 hPa | 34 (0) | AAO |
| Meridional winds 850 hPa | 33 (0) | SCA |
| Meridional winds 950 hPa | 32 (0) | SCA |













**Figures**

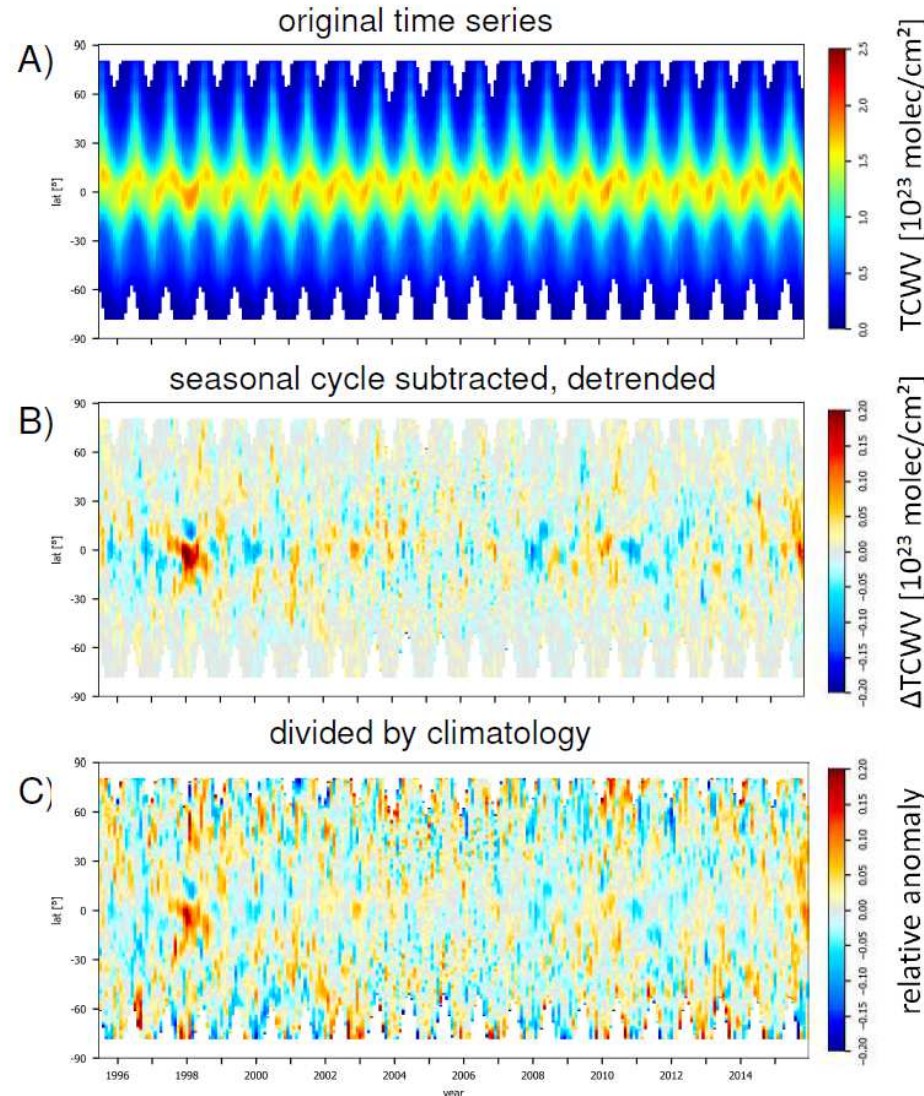


Fig. 1: A) TCWV measured from satellite as a function of time and latitude (zonally averaged values) on a 1° x 1° latitude/longitude grid with monthly resolution. B) (absolute anomalies) after the mean seasonal cycle and a linear trend was subtracted. C) relative anomalies (absolute anomalies divided by the corresponding monthly mean TCWV).








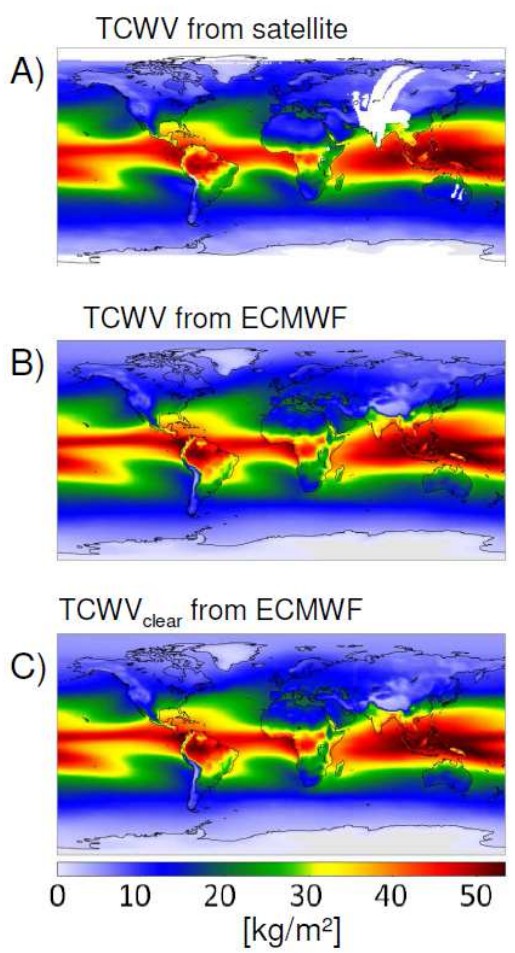

**Fig. 2: Global mean distribution of the TCWV from satellite observations (A) and ERA data: B) all data; C) only**
**clear sky observations during day.**









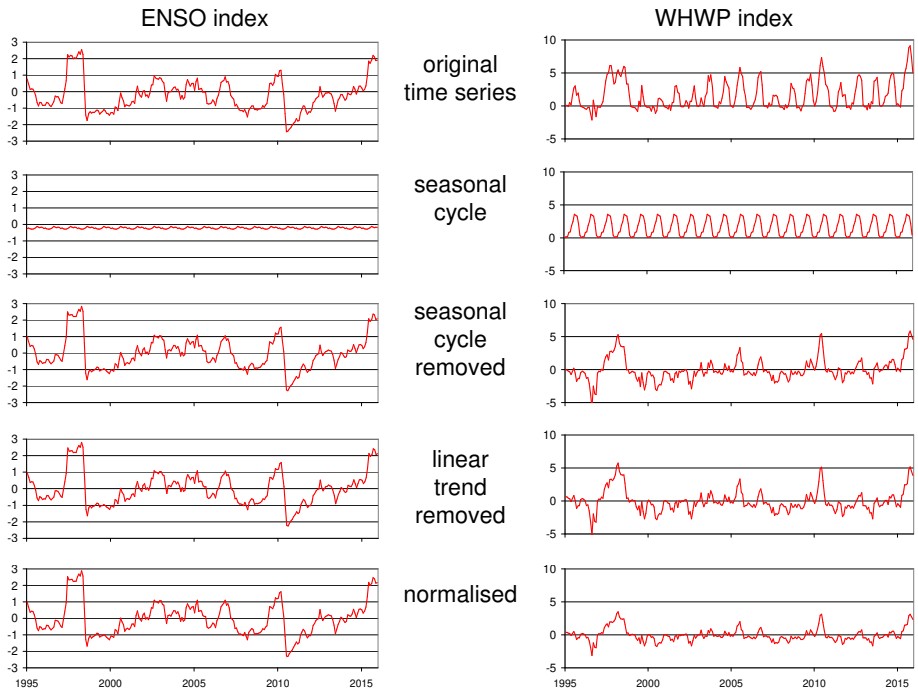

**Fig. 3: Illustration of the preparations for the indices before they are used in the fit to the global data sets: First, the mean seasonal cycles and linear trends are subtracted. Then the differences are normalised by their standard deviations.**





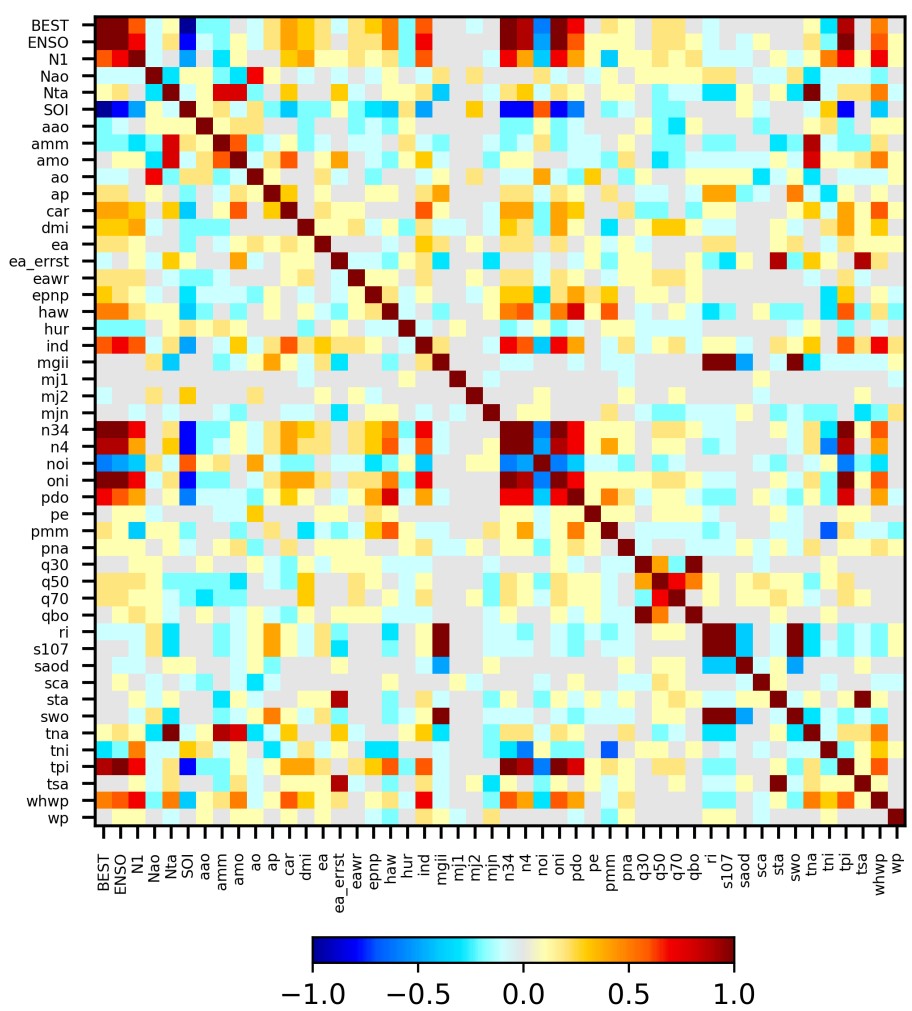


**Fig. 4: Correlation coefficients between the different teleconnection indices (after seasonal cycle was removed). Note that only one set of MJO indices is included here to minimise the total number of indices.**









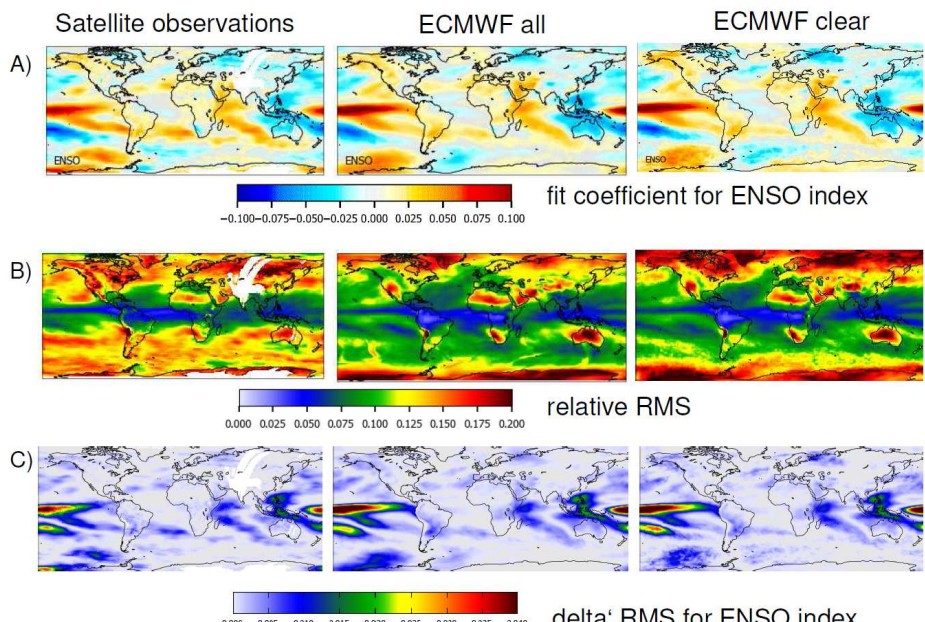

**Fig. 5: Global maps with the ENSO fit results for the three TCWV data sets. A) Fit coefficients; B) RMS of the differences between original data sets and fit functions; C) delta RMS values which describe the relative difference of the RMS if the ENSO index is included of excluded in the fit function (for details see text).**






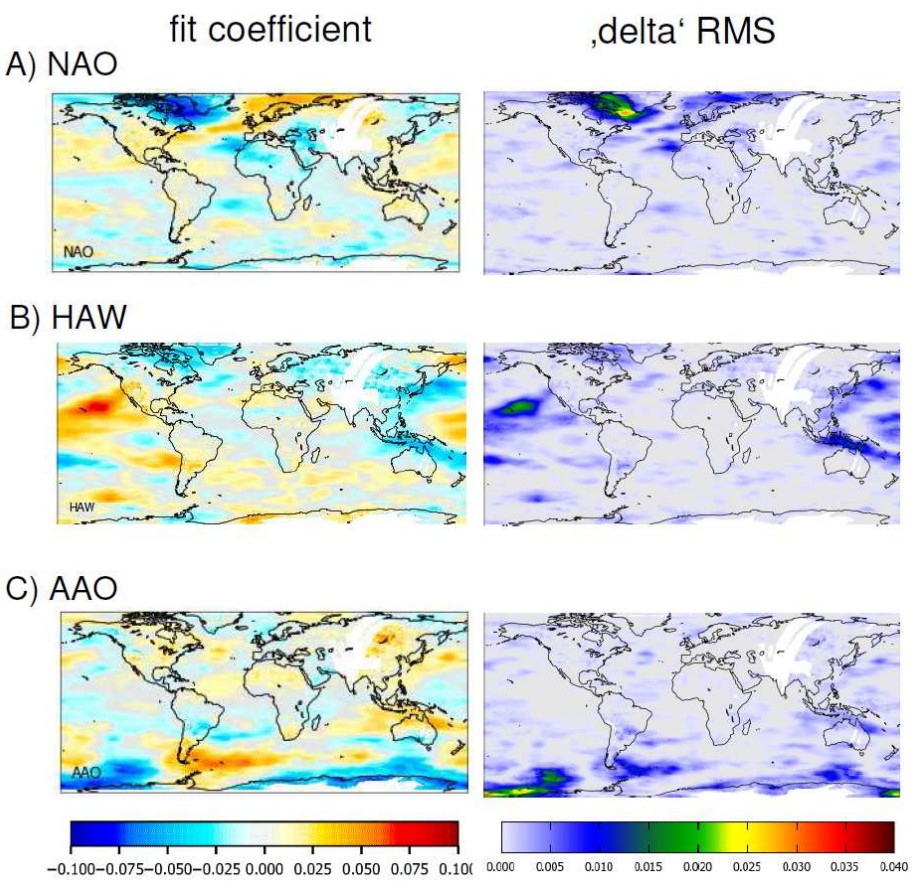

**Fig. 6: Global maps with the fit results (right) and delta RMS (left) for selected teleconnection indices with activity centers in northern high latitudes (A), Subtropics (B) and southern high latitudes (C). Results for the TCWV data set from satellite observations.**










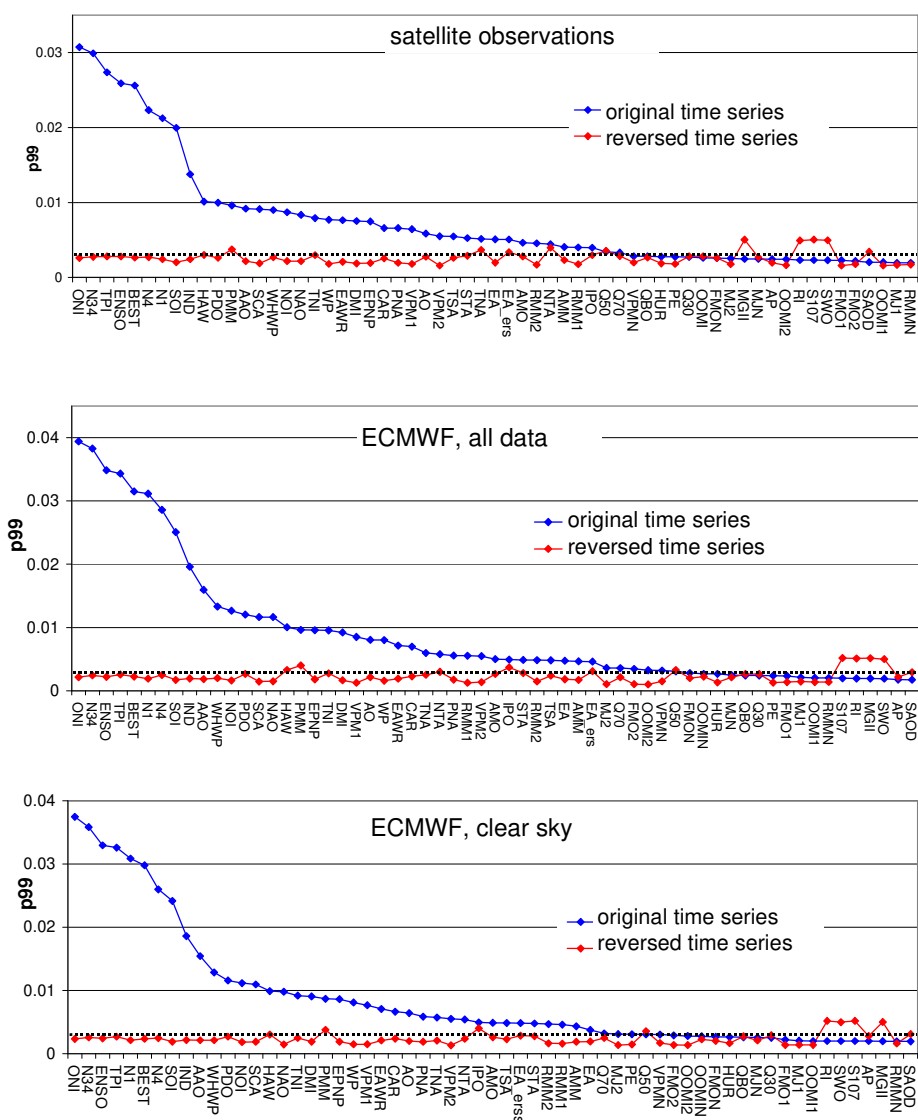


**Fig. 7: Blue markers: 99th percentiles (p99) of the delta RMS of the original indices for the three TCWV data sets. Red markers: similar results for the temporally reversed indices. Black lines: significance threshold. The indices are sorted from highest to lowest p99 values for the original indices.**




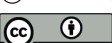

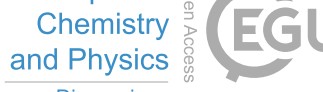

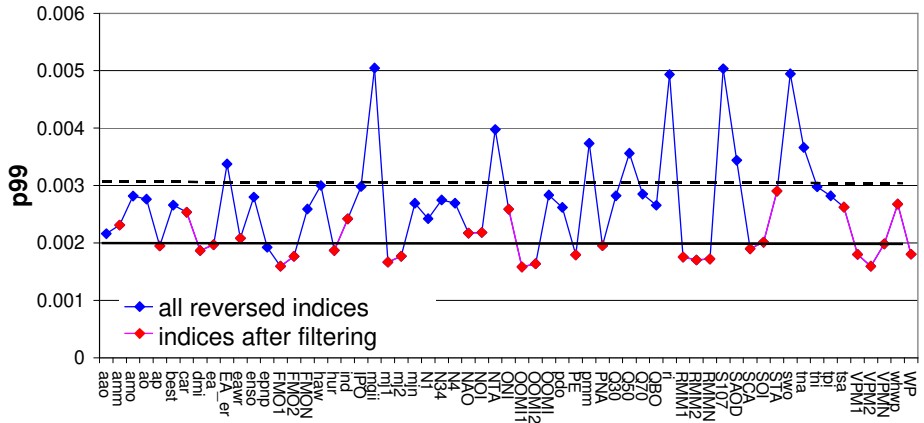

**Fig. 8: The 99th percentiles (p99) of the delta RMS of the temporally reversed indices for the TCWV from satellite observations (same as in Fig. 7, top). The blue markers indicate indices which are excluded from the calculation of the significance threshold (for details see text).**















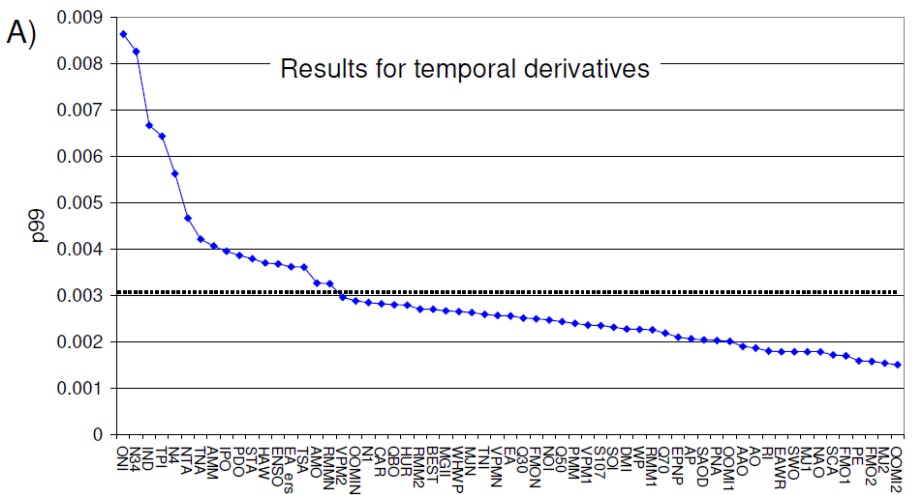

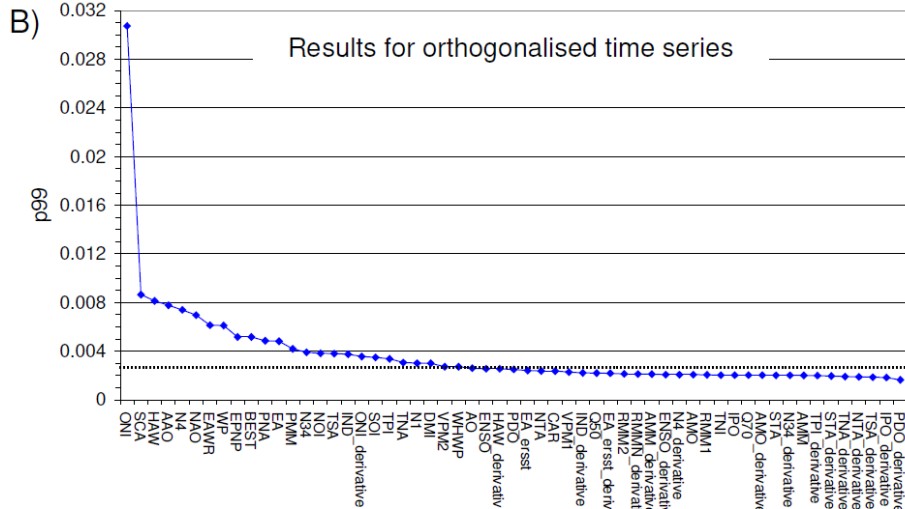

Fig. 9. The 99th percentiles (p99) of the delta RMS of the derivatives of the indices (A) and the orthogonalised indices (B). The black lines represent the significance threshold. The indices are sorted from highest to lowest p99 values.








A) Significant original indices satellite

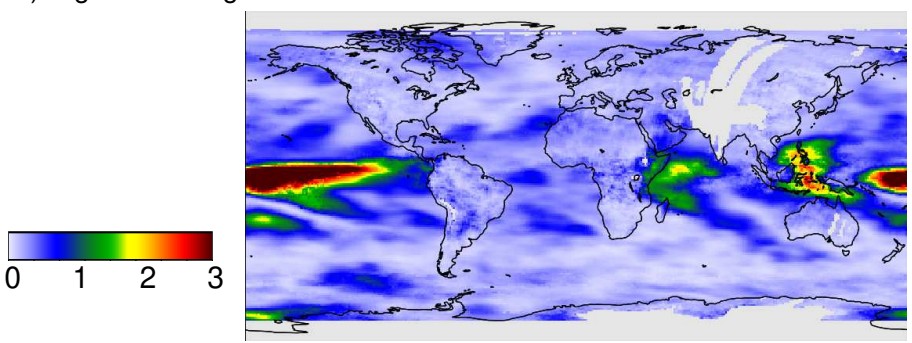

B) Significant orthogonalised indices satellite

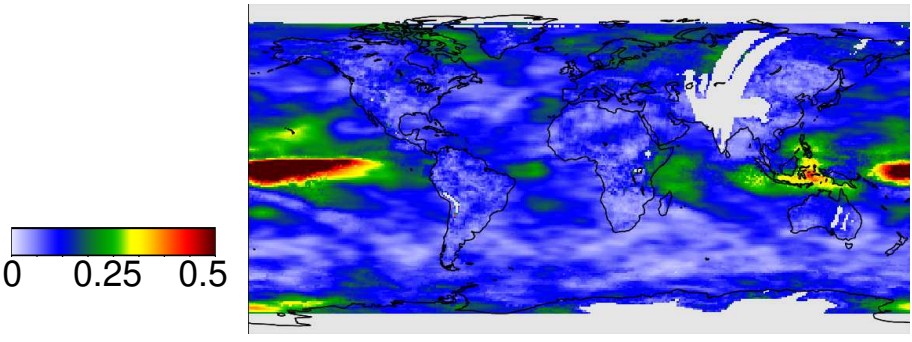

C) Significant orthogonalised indices ECMWF

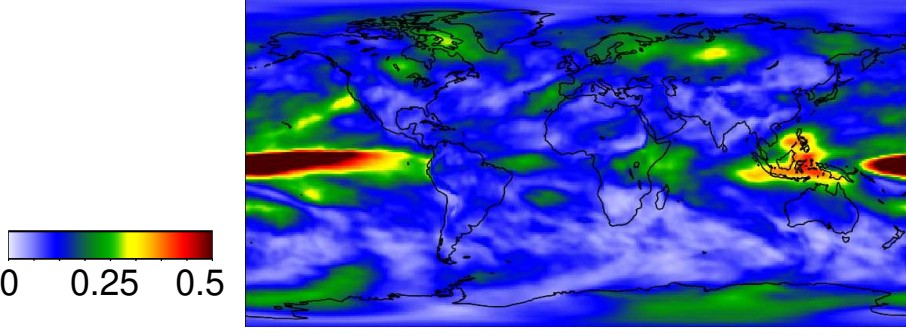


**Fig. 10: Cumulative delta RMS for different selections of indices and data sets (note the different colour scales).**








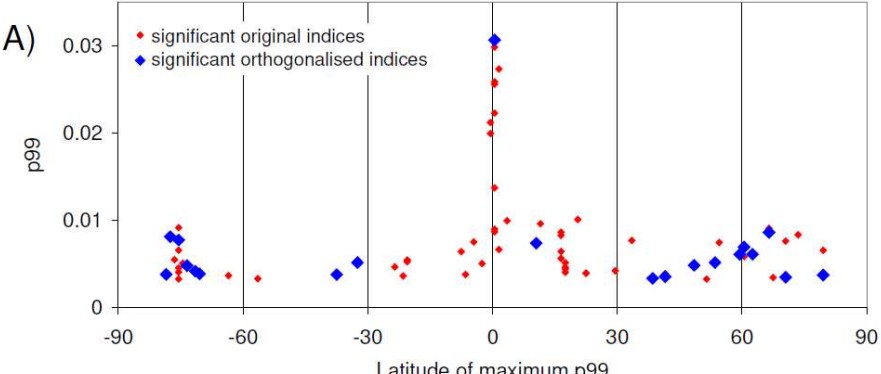

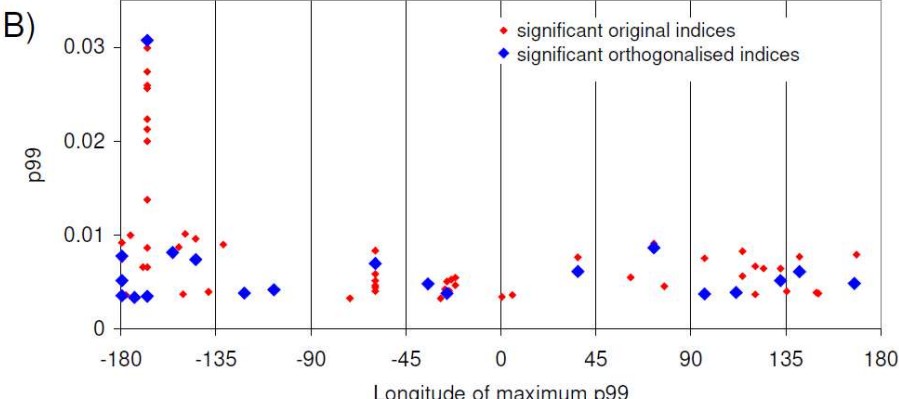


**Fig. 11: Location of the 99th percentile of the delta RMS values as function of latitude (A) or longitude (B). Red points indicate results for the original indices, blue points for the orthogonalised indices.**









**Appendix 1**

**Effect of the temporal correlation of the reversed indices with the original indices**

The 99th percentiles in Fig. 7 are substantially higher for several reversed indices than for others. Since all reversed indices represent non-geophysical variations, such enhanced 99th percentiles are not expected. Thus this finding was further investigated. It turned out that the enhanced values are caused by accidental correlations of these reversed indices with

original indices (see Fig. A6), for which high 99th percentile values are found. This reasoning is confirmed by the results shown in Fig. A7. There, high p99 values for reversed indices are always found if they are correlated with original indices with high p99 values. To avoid the effects of such accidental enhanced p99 values, only the reversed indices with no obvious correlations with original indices with high p99 values were kept for further processing (red boxes in Fig. A6). Here it should be noted that two somehow arbitrary choices were made:

a) the selection of the selected reversed indices (red boxes in Fig. A6) was made by visual inspection.

b) the influence of the correlation of the reversed indices with the original indices was only investigated for the 8 original indices with the highest p99 values.

Fortunately, both choices had only a minor influence on the derived threshold value. With respect to the first point, it should be noted that while the selection was made rather conservatively, still many reversed indices were kept after the filtering

process. It was also found that most of the skipped reversed indices were skipped because of enhanced correlations with several original indices. With respect to the second point it should be noted that it makes sense to consider only the original indices with the highest p99 values, because the correlations of the reversed indices with the original indices are in general rather low (see Fig. A6). The p99 values of the selected 8 original indices with the highest p99 values are in general substantially higher than the p99 values of the remaining indices. In sensitivity studies we found that taking account more

than 8 original indices had a negligible effect on the derived threshold values.

The red markers in Fig. 7 represent the p99 values for the indices which were kept after applying the selection criteria explained above. In the final step, from these p99 values the average and standard deviation are calculated. The p99 threshold for the significance of a indices is then calculated as the sum of the average plus three times the standard deviation (for the TCWV data set from satellite observations the threshold is: $0.00200 + 3*0.00036 = 0.00309$). This procedure was

chosen, because the threshold values calculated in this way are very close to the maximum p99 values of the remaining indices (red dots in Fig. 7) but are hardly affected by possible remaining outliers. The derived threshold value is indicated by the dashed black line in Fig. 7.









**Appendix 2**

**Additional Figures**

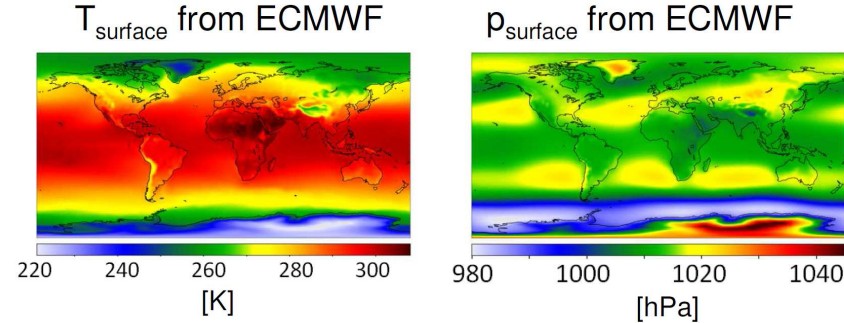

Fig. A1: Long-term mean distribution of the surface temperature (left) and extrapolated surface pressure (right)
from ECMWF reanalysis.










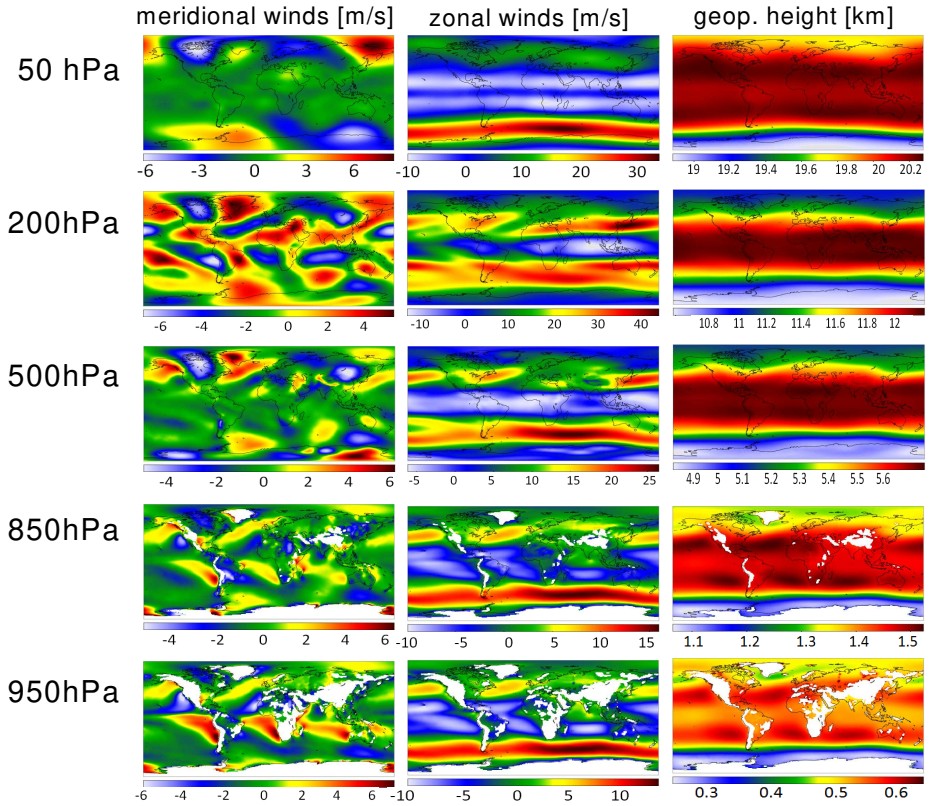

**Fig. A2: Long-term mean distribution of the meridional (left) and zonal (middle) winds as well as geopotential heights (right) at different pressure levels from ECMWF reanalysis.**


| Name (data source) | Short name | Type | Indices 1995 - 2015 |
|---|---|---|---|
| **A) Indices similar to ENSO** | | | |
| Multivariate ENSO Index (NOAA) | ENSO | Oceanic & Atmospheric | |
| Bivariate ENSO Timeseries (NOAA) | BEST | Oceanic & Atmospheric | |
| Oceanic Nino Index (NOAA) | ONI | Oceanic | |
| Nina 3.4 (NOAA) | N34 | Oceanic | |
| Nina 4 (NOAA) | N4 | Oceanic | |
| Tripole Index for the Interdecadal Pacific Oscillation (NOAA) | TPI | Oceanic | |
| Indian Ocean Index (NOAA) | IND | Oceanic | |
| | | | |
| B) Oceanic indices | | | |
| | | | |
| Nina 1 + 2 (NOAA) | N1 | Oceanic | |
| Pacific Decadal Oscillation (NOAA) | PDO | Oceanic | |
| Interdecadal Pacific Oscillation (Ministry of environment, NZ) | IPO | Oceanic | |

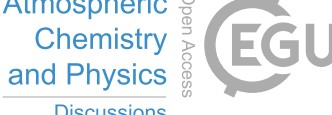

| Western Hemisphere warm pool (NOAA) | WHWP | Oceanic | |
|---|---|---|---|
| North Tropical Atlantic (NOAA) | NTA | Oceanic | |
| Tropical Northern Atlantic (NOAA) | TNA | Oceanic | |
| South Tropical Atlantic (NOAA) | STA | Oceanic | |
| Tropical Southern Atlantic (NOAA) | TSA | Oceanic | |
| Equatorial Atlantic Index (NOAA) | EA_ersst | Oceanic | |
| Caribbean Index (NOAA) | CAR | Oceanic | |
| Atlantic Meridional Mode (NOAA) | AMM | Oceanic | |
| Pacific Meridional mode (University of Wisconsin, USA) | PMM | Oceanic | |
| Atlantic multidecadal Oscillation (NOAA) | AMO | Oceanic | |
| Hawaiian Index (NOAA) | HAW | Oceanic | |
| Dipole Mode Index (NOAA) | DMI | Oceanic | |



| Trans-Nino index (NOAA) | TNI | Oceanic | |
|---|---|---|---|
| | | | |
| C) Atmospheric indices **(except MJO indices)** | | | |
| Southern Oscillation Index (NOAA) | SOI | Atmospheric | |
| Northern Oscillation Index (NOAA) | NOI | Atmospheric | |
| North Atlantic Oscillation (NOAA) | NAO | Atmospheric | |
| Pacific/North American pattern (NOAA) | PNA | Atmospheric | |
| East Atlantic pattern (NOAA) | EA | Atmospheric | |
| East Atlantic/Western Russia pattern (NOAA) | EAWR | Atmospheric | |
| Scandinavia pattern (NOAA) | SCA | Atmospheric | |
| West Pacific pattern (NOAA) | WP | Atmospheric | |
| East Pacific/North Pacific pattern (NOAA) | EPNP | Atmospheric | |
| Polar-Eurasian pattern (NOAA) | PE | Atmospheric | |

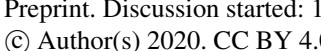

| Arctic Oscillation (NOAA) | AO | Atmospheric | |
|---|---|---|---|
| Antarctic Oscillation (NOAA) | AAO | Atmospheric | |
| Quasi-Biennial Oscillation at 30 hPa (NOAA) | QBO | Atmospheric | |
| Quasi-Biennial Oscillation at 30 hPa (Free University of Berlin) | Q30 | Atmospheric | |
| uasi-Biennial Oscillation at 50 hPa (Free University of Berlin) | Q50 | Atmospheric | |
| uasi-Biennial Oscillation at 70 hPa (Free University of Berlin) | Q70 | Atmospheric | |
| | | | |
| D) MJO Indices* | | | |
| Madden Julian Oscillation (OMI) Component 1 (NOAA) | MJ1 | Atmospheric | |
| Madden Julian Oscillation (OOMI) Component 1 (NOAA) | OOMI1 | Atmospheric | |
| Madden Julian Oscillation (FMO) Component 1 (NOAA) | FMO1 | Atmospheric | |
| Madden Julian Oscillation (VPM) Component 1 (NOAA) | VPM1 | Atmospheric | |
| Madden Julian Oscillation** Component 1 (Australian Bureau of Meteorology) | RMM1 | Atmospheric | |



| | | | |
|---|---|---|---|
| Madden Julian Oscillation (OMI) Component 2 (NOAA) | MJ2 | Atmospheric | |
| Madden Julian Oscillation (OOMI) Component 2 (NOAA) | OOMI2 | Atmospheric | |
| Madden Julian Oscillation (FMO) Component 2 (NOAA) | FMO2 | Atmospheric | |
| Madden Julian Oscillation (VPM) Component 2 (NOAA) | VPM2 | Atmospheric | |
| Madden Julian Oscillation** Component 2 (Australian Bureau of Meteorology) | RMM2 | Atmospheric | |
| Madden Julian Oscillation (OMI) Sum of both compoenents (NOAA) | MJN | Atmospheric | |
| Madden Julian Oscillation (OOMI) Sum of both (NOAA) | OOMIN | Atmospheric | |
| Madden Julian Oscillation (FMO) Sum of both (NOAA) | FMON | Atmospheric | |
| Madden Julian Oscillation (VPM) Sum of both (NOAA) | VPMN | Atmospheric | |
| Madden Julian Oscillation** Sum of both (Australian Bureau of Meteorology) | RMMN | Atmospheric | |
| | | | |
| E) Other Indices | | | |
| Composite MG II index (University of Bremen) | MGII | solar | |





| Magnetic AP index (NOAA, GeoForschungsZentrum, Postdam) | AP | solar | |
| --- | --- | --- | --- |
| Radio flux at 10.7 cm (NOAA, Penticton, B.C., Canada) | S107 | solar | |
| Sun spot number (NOAA, SWPC Space Weather Operations) | SWO | solar | |
| Sun spot number (NOAA, S.I.D.C. Brussels International Sunspot Number) | RI | solar | |
| **Stratospheric AOD***** **(LATMOS/IPSL)** | SAOD | Atmospheric | |
| **Total number of hurricanes in Atlantic region (NOAA)** | HUR | Atmospheric | |

**Fig. A3: List of original indices used in this study. Besides the short names also the data sources and figures with their variation from 1995 to 2016 are shown.**

\*All MJO indices are convoluted with a Gaussian kernel fo 30 days FWHM; \*\*Original index according to Wheeler and Hendon, 2004;
\*\*\*Khaykin et al., 2017













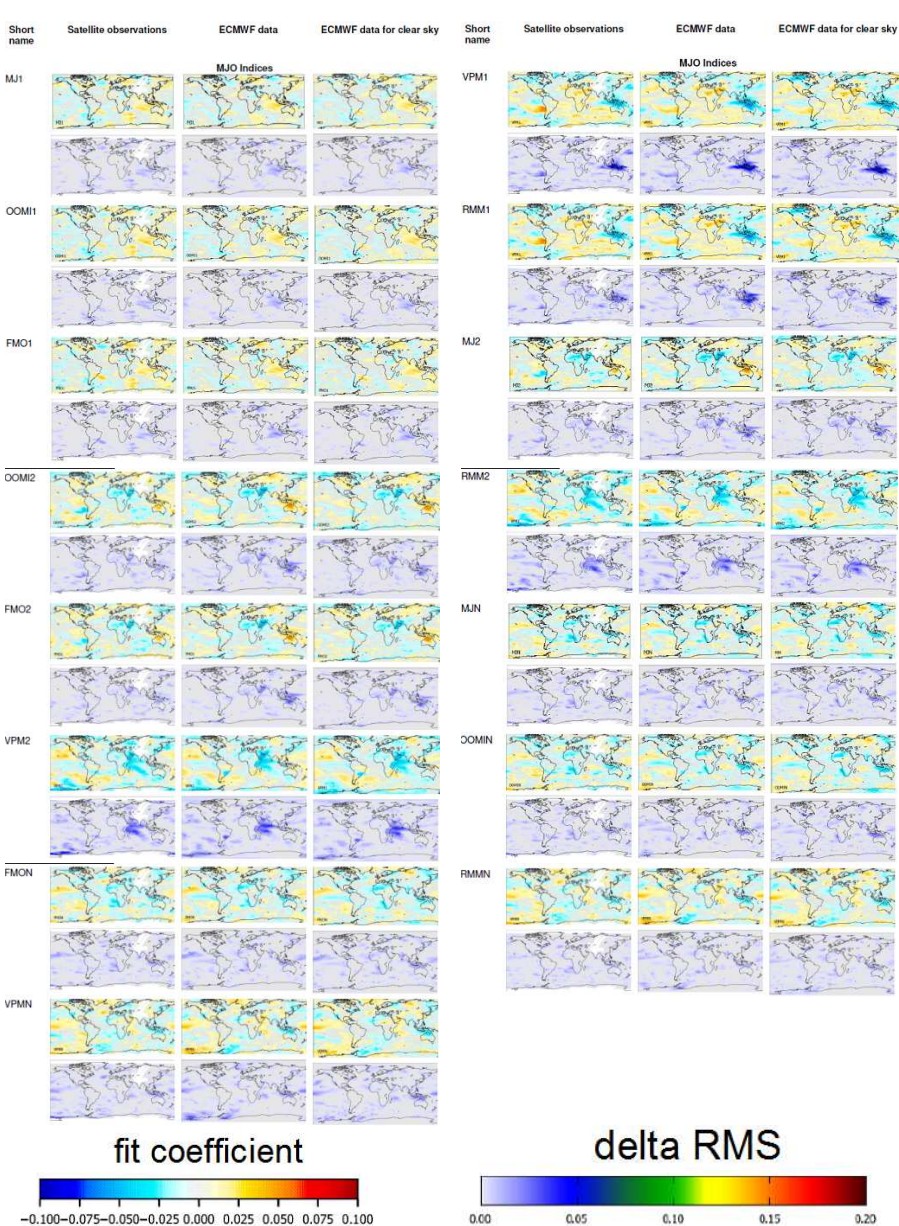





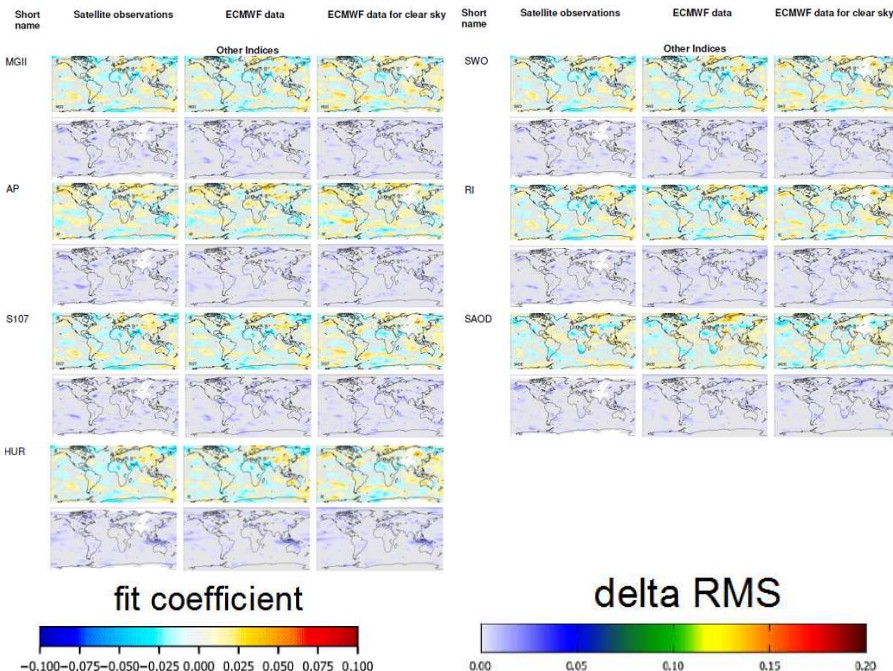

**Fig. A4: Fit coefficients (top) and delta RMS values (bottom) for all indices used in this study. Shown are the results for the three water vapor data sets: satellite observations (left), model results (center), and model results for clear sky conditions (right).**





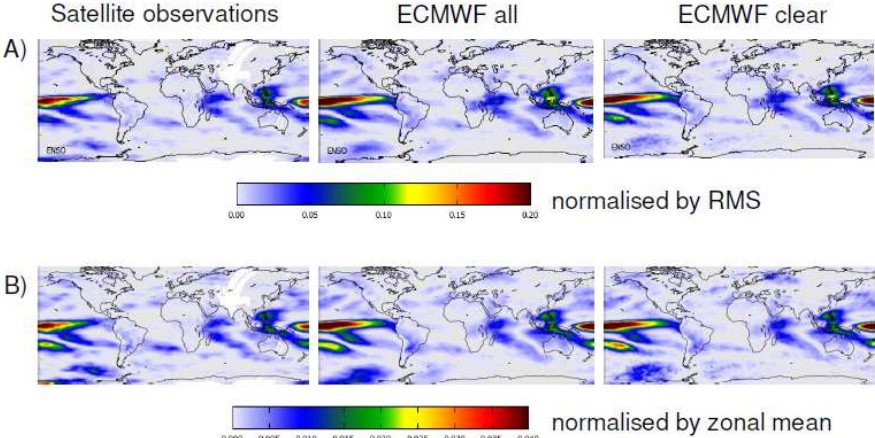

**Fig. A5: Comparison of delta RMS values for the ENSO index calculated in two different ways. A) The difference of the RMS with and without the ENSO index included in the fit is divided by the respective RMS of each 1°x1° pixel; B) The difference of the RMS with and without the ENSO index included in the fit is divided by the zonal mean of the TCWV at the same latitude. Note the different colour scales.**





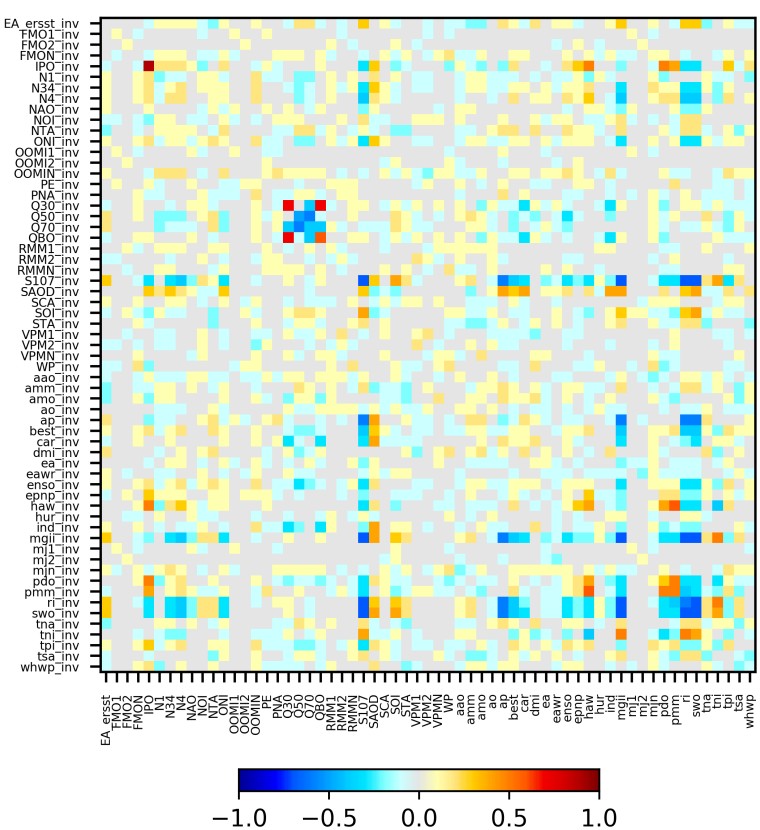

**Fig. A6: Correlation coefficients between the temporally reversed and original indices. For several combinations**
**enhanced coincidental correlations are found.**










**Fig. A7: Correlation plots for the 8 original indices with the highest p99 values. The blue dots represent the 61 reversed indices. The x-axis describes the correlation coefficients of the reversed indices with the selected original indices. The y-axis describes the p99 value or the reversed indices. High p99 values are found for the reversed indices which show high correlation to the original indices.**





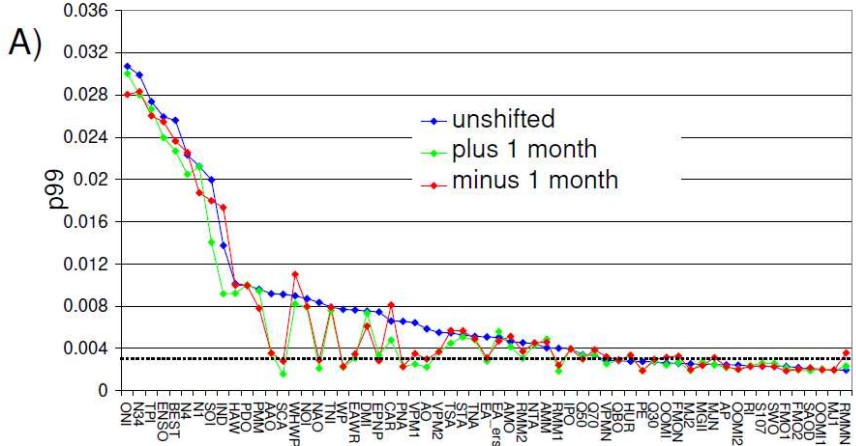

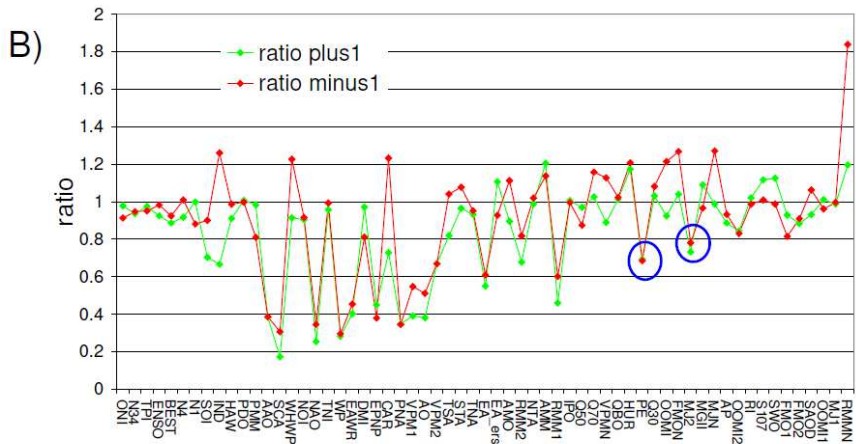


**Fig. A8: Top: 99th percentiles (p99) of the delta RMS values for the original (blue) and shifted indices (green: plus 1 month; red: minus 1 month). The indices are sorted from highest to lowest p99 values for the unshifted original indices. Bottom: ratios of the p99 values of the shifted and original indices. Results are for the TCWV data set from satellite observations. The blue circles indicate teleconnection indices with p99 values below the threshold, but ratios**
**of the shifted indices < 0.8.**








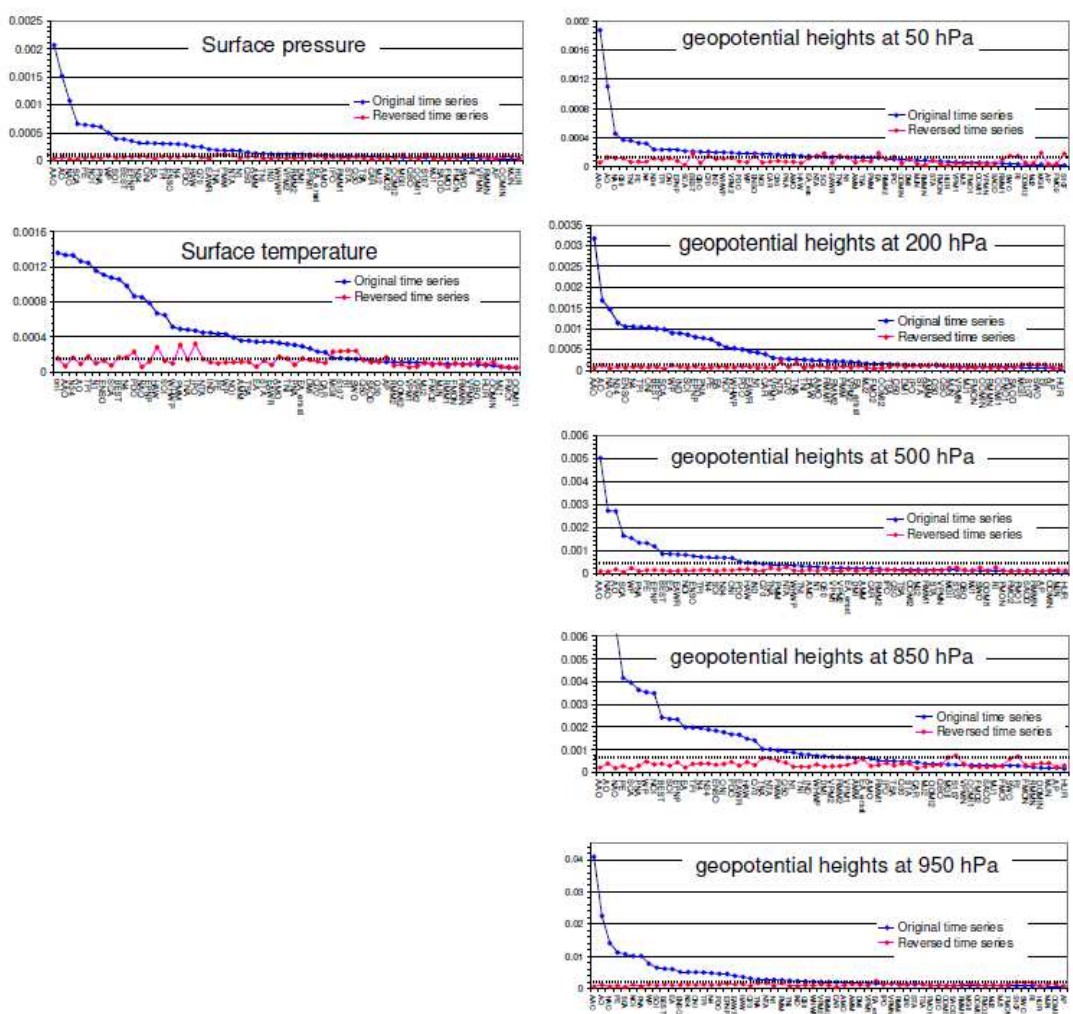






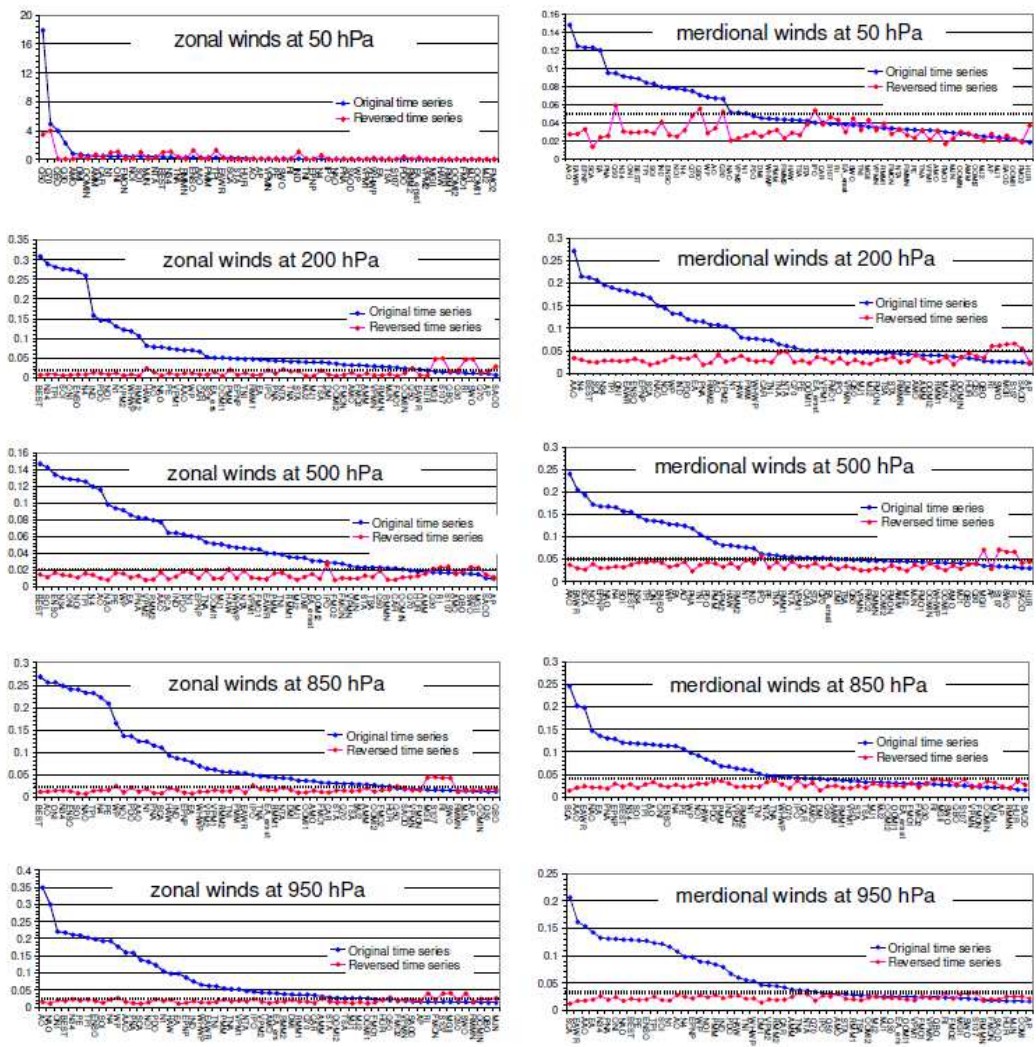

**Fig. A9: 99th percentiles of the delta RMS values (p99) found for the different indices in different global data sets.**
**Blue markers: p99 for the original indices; red markers: p99 for the temporally reversed indices; black lines: significance thresholds. The indices are sorted from highest to lowest p99 values for the original indices.**








| | Original indices | Orthogonalised indices | | Original indices | Orthogonalised indices |
|---|---|---|---|---|---|
| 1 | ONI | | 11 | PNA | PNA ortho |
| 2 | SCA | SCA ortho | 12 | EA | EA ortho |
| 3 | HAW | HAW ortho | 13 | PMM | PMM ortho |
| 4 | AAO | AAO ortho | 14 | N34 | N34 ortho |
| 5 | N4 | N4 ortho | 15 | NOI | NOI ortho |
| 6 | NAO | NAO ortho | 16 | TSA | TSA ortho |
| 7 | EAWR | EAWR ortho | 17 | IND | IND ortho |
| 8 | WP | WP ortho | 18 | ONI derivative | ONI derivative ortho |
| 9 | EPNP | EPNP ortho | 19 | SOI | SOI ortho |
| 10 | BEST | BEST ortho | 20 | TPI | TPI ortho |

**Fig. A10: Delta RMS maps for the significant orthogonalised indices together with the delta RMS maps for the original indices. The numbers at the left sides indicate the order (descending) of the p99 values (see also Fig. 8).**




**Table A1 Significant indices for all data sets (indices with p99 values below threshold but shift ratios <0.8 are indicated in brackets).**

| Data set | Number of significant indices | Significant indices (from highest to lowest p99 values) |
|---|---|---|
| TCWV sat | 42 (2) | ONI, N34, TPI, ENSO, BEST, N4, N1, SOI, IND, HAW, PDO, PMM, AAO, SCA, WHWP, NOI, NAO, TNI, WP, EAWR, DMI, EPNP, CAR, PNA, VPM1, AO, VPM2, TSA, STA, TNA, EA, EA_ersst, AMO, RMM2, NTA, AMM, RMM1, IPO, Q50, Q70 (PE, MJ2) |
| TCWV ERA | 44 (1) | ONI, N34, ENSO, TPI, BEST, N1, N4, SOI, IND, AAO, WHWP, NOI, PDO, SCA, NAO, HAW, PMM, EPNP, TNI, DMI, VPM1, AO, WP, EAWR, CAR, TNA, NTA, PNA, RMM1, VPM2, AMO, IPO, STA, RMM2, TSA, EA, AMM, EA_ersst, MJ2, Q70, FMO2, OOMI2, VPMN, Q50 (PE) |
| TCWV ERA clear | 42 (3) | ONI, N34, ENSO, TPI, N1, BEST, N4, SOI, IND, AAO, WHWP, PDO, NOI, SCA, HAW, NAO, TNI, DMI, PMM, EPNP, WP, VPM1, EAWR, CAR, AO, PNA, TNA, VPM2, NTA, IPO, AMO, TSA, EA_ersst, STA, RMM2, RMM1, AMM, EA (PE, FMO2, OOMI2) |
| Tsurf | 37 (1) | ONI, AAO, N34, AO, TPI, N1, ENSO, SCA, BEST, N4, PDO, NAO, EPNP, HAW, SOI, WHWP, PMM, TNA, IPO, NTA, IND, PE, WP, NOI, AMM, TSA, EA, STA, EAWR, AMO, TNI, PNA, EA_ersst, DMI, Q70, CAR (RMM2) |
| Spred | 35 (1) | AAO, AO, NAO, SCA, PE, NOI, PNA, WP, SOI, BEST, EPNP, N34, ONI, EA, TPI, ENSO, N4, PDO, HAW, Q70, EAWR, TNA, PMM, NTA, N1, Q50, AMM, TNI, IND, WHWP, VPM2, RMM2, DMI, VPM1 (RMM1) |
| Geopot 50 hPa | 17 (5) | AAO, AO, NAO, Q50, TNI, PE, N4, N34, TPI, ONI, EPNP (VPM2, PNA, EA, RMM2, RMMN) |
| Geopot 200 hPa | 40 (0) | AAO, AO, NAO, N34, ENSO, N4, TPI, ONI, BEST, SCA, WP, IND, SOI, EPNP, PNA, PE, EA, NOI, WHWP, PDO, EAWR, N1, CAR, VPM1, NTA, Q70, TNA, TNI, HAW, AMO, RMM1, RMM2, PMM, VPM2, EA_ersst, MJ2, FMO2, OOMI2, TSA |
| Geopot 500 hPa | 32 (1) | AAO, NAO, AO, SCA, WP, PNA, PE, EPNP, BEST, EA, EAWR, NOI, ENSO, TPI, N4, SOI, N34, ONI, PDO, HAW, IND, Q70, TNA, PMM, NTA, WHWP, TNI, AMO, N1, Q50 (RMM2) |
| Geopot 850 hPa | 33 (1) | AAO, AO, NAO, PE, SCA, PNA, WP, NOI, BEST, SOI, EPNP, EA, TPI, N4, N34, ENSO, ONI, PDO, EAWR, HAW, Q70, TNA, NTA, PMM, Q50, N1, TNI, IND, WHWP, DMI, VPM2 (RMM1) |
| Geopot 950 hPa | 30 (1) | AAO, AO, NAO, PE, SCA, NOI, PNA, WP, SOI, BEST, EA, ENSO, N34, ONI, TPI, N4, PDO, EPNP, EAWR, HAW, Q70, TNA, NTA, N1, PMM, TNI, IND, Q50 (RMM2) |
| Zonal winds 200 hPa | 51 (0) | BEST, N34, TPI, SOI, ONI, ENSO, N4, IND, PDO, NOI, N1, VPM2, WHWP, RMM2, HAW, AO, NAO, PE, VPM1, AAO, WP, CAR, SCA, EA_ersst, OOMI1, PMM, EPNP, TNI, RMM1, EA, IPO, PNA, NTA, TNA, STA, MJ2, MJ1, TSA, DMI, OOMI2, FMON, AMO, FMO2, AMM, VPMN, RMMN, MJN, FMO1, OOMIN, Q50 |
| Zonal winds 500 hPa | 49 (0) | BEST, SOI, ENSO, N34, ONI, NOI, TPI, N4, AO, NAO, PE, WP, EA, PNA, VPM2, RMM2, AAO, SCA, IND, PDO, N1, EPNP, TNA, OOMI1, MJ1, HAW, WHWP, NTA, VPM1, FMO1, EAWR, PMM, TNI, RMM1, MJ2, DMI, EA_ersst, OOMI2, IPO, FMO2, FMON, VPMN, MJN, STA, TSA, Q50, RMMN, CAR, OOMIN |
| Zonal winds 850 hPa | 46 (1) | BEST, AO, ONI, N34, ENSO, SOI, NAO, TPI, N4, PE, WP, NOI, PDO, AAO, N1, PNA, SCA, HAW, IND, EPNP, EA, WHWP, VPM2, VPM1, RMM2, TNI, PMM, EAWR, IPO, TNA, EA_ersst, RMM1, NTA, DMI, MJ1, OOMI1, AMO, FMO1, CAR, STA, Q70, TSA, MJ2, AMM, OOMI2 (FMO2) |
| Zonal winds 950 hPa | 42 (4) | AO, NAO, ONI, BEST, N34, PE, TPI, ENSO, SOI, N4, WP, AAO, PNA, SCA, NOI, PDO, N1, HAW, EA, EPNP, IND, |





| | | WHWP, EAWR, TNI, PMM, TNA, VPM1, NTA, IPO, VPM2, AMO, EA_ersst, RMM2, DMI, RMM1, Q70, CAR, AMM (OOMI2, MJ1, MJ2, OOMI1) |
|---|---|---|
| Meridional winds 50 hPa | 24 (3) | AAO, EAWR, EPNP, SCA, EA, PNA, Q50, N34, ONI, BEST, TPI, SOI, IND, ENSO, NOI, N4, Q70, QBO, WP, AO (VPM2, RMM2, PE) |
| Meridional winds 200 hPa | 32 (1) | AAO, N4, BEST, SOI, N34, TPI, ONI, EAWR, ENSO, EPNP, SCA, NAO, NOI, WP, IND, PDO, EA, PNA, RMM2, AO, VPM2, N1, HAW, PMM, WHWP, CAR, TNI, TNA, NTA, Q70 (FMO1) |
| Meridional winds 500 hPa | 34 (0) | AAO, EAWR, SCA, NOI, EPNP, NAO, N4, SOI, BEST, N34, TPI, ONI, ENSO, WP, EA, AO, PNA, TNI, PDO, PMM, VPM2, HAW, RMM2, N1, IND, IPO, PE, TNA, RMM1, NTA, VPM1, CAR, STA |
| Meridional winds 850 hPa | 33 (0) | SCA, AAO, EAWR, NAO, EA, PNA, EPNP, BEST, N34, SOI, TPI, AO, ONI, ENSO, N4, PE, WP, NOI, HAW, PDO, PMM, IND, VPM2, RMM2, N1, TNI, NTA, TNA, WHWP, Q70, IPO, CAR |
| Meridional winds 950 hPa | 32 (0) | SCA, EAWR, AAO, EA, N34, PNA, ONI, NAO, BEST, PE, ENSO, TPI, SOI, N1, AO, N4, EPNP, WP, NOI, PMM, IND, PDO, HAW, TNI, WHWP, DMI, VPM2, RMM2, CAR, AMM, TNA |
