# Peer review of "Identification of atmospheric and oceanic teleconnection patterns in a 20-year global data set of the atmospheric water vapor column measured from satellites in the red spectral range"

_Atmospheric Chemistry and Physics, 2020_

## Referee Comment (RC1) · Anonymous Referee #2 · 30 Sep 2020

**Review of ACP-2020-565**

**Recommendation:**

General comments: The authors are attempting to show how total column water vapor (TCWV) can be used to reveal the presence of atmospheric teleconnections seen in other datasets. This method is certainly interesting and could be of value, at least in the context of demonstrating the utility of TCWV in revealing existing teleconnections. However, the presentation in this paper was extremely difficult to follow as the authors jumped from one analysis to another with no clear direction as to why. There were many different technical approaches employed within this study, and while these likely have value in the context of what the authors' research goals are, the reasons for using the methods they employ were not well established. Further, the authors state early in the study that they are going to compare the results with similar results from pressure, temperature, etc. fields more traditionally utilized in teleconnection studies. I did not see these comparisons. In general, the authors focused too heavily on the significance of the relationship between their empirical estimates of the TCWV using the teleconnection index and the TCWV itself. It read more like a study attempting to predict monthly TCWV using teleconnections, not a study linking TCWV to teleconnections. Either the study should be reframed in that context or the authors need to do a better job of linking their results back to the teleconnections they are trying to predict. Which teleconnections were predicted well? Which were predicted poorly? Why? Such discussion was absent from this study and seems directly relevant to the research objectives outlined therein.

**Specific comments:**

Most of the work done in PCA-based teleconnection studies in pressure/geopotential height is confined to midlatitude and Arctic regions in the Northern Hemisphere owing to the barotropic conditions in the tropical latitudes. This should be better specified by the authors.

If multiple indices characterizing the same phenomena exist (e.g. MJO, ENSO), why include them all? How do you reconcile the differences in how those indices are characterizing their teleconnection and relate those differences back to your results? (Lines 135-137).

In the fit functions, how were the quantities *c* and *b* determined? Were they based on a fit with the satellite data, the ERA, etc.? Nothing is provided in the text in this regard.

The authors discuss the use of "reversed datasets" in section 5.1. However, they provide no discussion of what was reversed. Was it just the teleconnection time series? Was it the TCWV time series? Were they reversed in time? Did you just reverse the index numbers directly, as is done frequently in pattern recognition and database type work? I don't see why, if the reverse was temporal, why the correlations didn't simply change sign but remain the same magnitude. The authors need to provide a lot more explanation on this aspect of their study as they do not really describe it in much detail. Why did you do this?

The main crux of what the authors were doing here was attempting to show significance of fit between teleconnection indices and the water vapor datasets they employ. However, while the fits may be "significant", how much of that is a result of sample size and how much is a result of the true quality of fit? In other words, how "good" are those fits? What are the R2 values for those fits? A model can have

a significant fit with a very low variance explained result if the sample size is large enough. It is unclear how the "significance" of the results tie into the quality of the fit and the RMS values themselves.

In section 8 the authors state they "orthogonalized" their indices. What method was used to do this? Why did they do this?

Technical corrections:

The e.g. on line 51 can be removed.

What is a "time series like index"? (Line 78)

In Figures A1 and A2, are the times over which these averages were computed the same 1995-2015 time period? The ERA have a longer period of record so it would be good to specify this.

It is not clear why Figure A3 is included in the text. There are too many time series and their individual value in the study is not clear.

Figure A4 is almost impossible to read. There should be a compelling reason why this figure is included in the text as it includes well over 200 maps. The authors should choose which of those figures best illustrate their point and include those instead of including them all.

---

## Referee Comment (RC2) · Anonymous Referee #1 · 10 Oct 2020

The manuscript selects extensive existing teleconnection indices and aims to identify teleconnection patterns in a new global dataset. It presents a new method to examine the reproducibility of the teleconnection in the global dataset along with other data sources. Although the manuscript first employs the dataset to the teleconnection research, which shows its novelty, the quality of the presentation needs substantial improvements. In the current version, the manuscript intends to address the research questions mentioned in the introduction, but the presentation of the results is confusing and difficult to follow. For example, the manuscript (e.g., Page 2, line 66-67) frequently

mentioned the aim is to investigate the influence of teleconnections on the global distribution of the total column water vapor (TCWV). As far as I understand, the paper does not clearly address this issue. Could the authors clarify and stress the influence of teleconnection on the TCMV in the manuscript?

Hence, I suggest some restructure of the body text of the manuscript. For instance, it might be beneficial to clarify the relationship between different groups of indices and their corresponding results. Also, I suggest the authors improve the quality of the figures.

Specific comments:

1. The manuscript used the water vapor column data from satellite observations in the red spectral range. Is there anything special for the use of red spectral range in the paper? It will be good to give some explanations otherwise I suggest removing "in the red spectral range" from the title. 2. Page 3, line 105: Could the authors provide one or two references, which shows that the variations of the TCWV are strongly associated with ENSO events? Or can authors provide the correlations over the tropical band? 3. In the third section (Page 4, line 130-145), the authors did a great piece of work on putting various existing teleconnection indices together. The manuscript divides those indices into groups but indices in the same group can have high correlations, like ENSO indices (Fig. A3). The authors could focus on some selected indices and omit other highly correlated indices unless the differences among those indices affect the conclusion of the manuscript. It would be good to see more discussions in the line 143-144 for Fig.4. 4. In the fifth section, the manuscript uses the reversed indices but a clear explanation on the reversed indices and its meaning is needed, e.g. what is the meaning of 'reversed'. The current presentation makes Fig. 7 hard to understand. 5. Page 7, line 289-292: the results here are interesting. Could the authors provide more physical or dynamical explanations behind these results?

Minor comments: 1. Page 3, line 116-117: Could the authors add more descriptions

for Fig. 2? 2. Page 3, line 125: I suggest removing Figs. A1 and A2. 3. Page 4, line 138: The manuscript used the word 'fit' here, but the introduction of the fit function is shown in the next section. Moving the description of the fit function here might improve the clarity. Section 3 and 4 should be organized in a logical flow. 4. Page 5, line 174: Could the authors clarify the meaning of a larger or lower value of delta RMS (eq. 3)? 5. Table 2 and Figure 5 captions.

Please also note the supplement to this comment:
https://acp.copernicus.org/preprints/acp-2020-565/acp-2020-565-RC2-supplement.pdf

---

## Author Comment (AC1) · 13 Dec 2020

Response to Reviewer 1

We thank the reviewer for the constructive and helpful remarks suggestions. We followed most of these suggestions as described below. Before we give our detailed answers to the individual comments, we first summary the most important changes to the manuscript.

The following major changes were made to the revised version of the manuscript:

**A) The motivation and description of the empirical method was improved.**

We agree with both reviewers that the description of new method had errors and was complicated. It was also not well motivated.

In the revised version we shifted many technical details of the new method to the appendix (e.g. the description of our normalisation approach as new appendix A1, or the investigation of the effect of time shifts as new appendix A3).

We also added more details about the fit function to section 4, and more details about the calculation of the temporally reversed indices to section 5.1.

For the motivation of our new approach we added the following information to section 5 (see also new Fig. 6):

[revised manuscript text omitted]

**C) The relationships between different indices and the motivation for the orthogonalisation of the indices was made more clear:**

We added new columns in Table 2 (see below). We now show separate columns for indices similar to ENSO, polar atmospheric indices, MJO indices, as well as other oceanic and atmospheric indices.

| Indices similar to ENSO (7) | Other oceanic indices (16) | Atmospheric polar indices (8) | MJO indices (15) | Other atmospheric indices (8) | Others indices (7) |
|---|---|---|---|---|---|
|  |  |  |  |  |  |

| | | | | | |
|---|---|---|---|---|---|
| BEST | HAW | SCA | MJ1 | PNA | Solar indices: |
| N34 | PDO | AAO | MJ2 | SOI | RI |
| TPI | PMM | EAWR | MJN | NOI | MGII |
| ONI | N1 | NAO | VPM1 | EA | SWO |
| ENSO | TNI | EPNP | VPM2 | QBO | S107 |
| N4 | NTA | AO | VPMN | Q30 | AP |
| IND | TNA | PE | RMM1 | Q50 | |
| | WHWP | WP | RMM2 | Q70 | HUR |
| | IPO | | RMMN | | (hurricane |
| | CAR | | OOMI1 | | frequency) |
| | AMO | | OOMI2 | | |
| | DMI | | OOMIN | | SAOD |
| | AMM | | FMO1 | | (stratospheric |
| | STA | | FMO2 | | AOD) |
| | TSA | | FMON | | |
| | EA_ersst | | | | |

In section 3 we added the following explanation:
' Many of these indices (describing the same phenomenon), but also many of the other teleconnection indices are highly correlated. The strength of these correlations is presented in Fig. 3 as a matrix with correlation coefficients between the different indices (after the seasonal cycles were removed). In spite of the correlations amongst the teleconnection indices, we decided as a first step to include them all in our study, because beforehand it is not clear which index might be best suited to represent a teleconnection phenomenon. Using our empirical approach, however, it becomes possible to quantify the significance and strength of the different indices and thus to select the best suited index for a given teleconnection phenomenon. Finally, we apply an orthogonalisation for the most significant indices (see section 7) to minimise the effect of the correlations and to identify the dominant temporal teleconnection patterns in our TCWV data set.'

To better motivate the orthogonalisation, we modified the respective information in section 7 to:
, To account for correlations between the different indices, we thus applied an orthogonalisation approach. For the orthogonalisation (based on the Gram–Schmidt process), all ‚significant' original indices and significant temporal derivatives (see Figure A11) were considered (in total 57 indices). The order of indices used in the iterative orthogonalisation process was from highest to lowest p99 values. The result of the orthogonalisation approach is a set of modified teleconnection indices, which shows zero correlation amongst each other (for the considered time period). Thus this new set of orthogonalised indices can be used to determine the number of independent significant teleconnection patterns in the global water vapor data sets. We applied our new method to the new set of orthogonalised indices to test which of the modified indices have p99 values above the significance threshold.'

**D) The logical flow of the paper and the appearance was improved.**

As mentioned above several technical parts were shifted to the appendix. The science questions were better motivated in the introduction, and the corresponding answers were added to the conclusions.
Several Figures were shifted/deleted/modified:
-Fig. 3 was shifted to the appendix
-Fig. 8 was shifted to the appendix
-the upper part of Fig. 9 was shifted to the appendix
-Figs A1 and A2 were deleted as suggested
-the quality of Fig. A4 was improved and the number of the sub-figures was largely reduced (by a factor of 3)
-the quality of Fig. A9 was improved

**E) We added a new sub-section (6.1) for the comparison of the spatial patterns of the measured and simulated TCWV.**
While for most teleconnection indices very good agreement of the spatial patterns is found between the measuremed and simulated TCWV, for some indices also substantial differences are detected. These differences can point to shortcomings in either the satellite or model data sets (or both) and might be helpful for corresponding improvements.
We added a new Fig. 8 (see below) and the following new text:
'For most of the teleconnection indices, very similar spatial patterns are found in the TCWV data sets obtained from satellite or ECMWF data (see Fig. A9). This confirms both the high quality of the satellite measurements and model simulations. However, for some indices, also substantial differences are found (see Fig. 8). The most obvious differences are found over northern Africa. In principle, they could be caused by errors of both the satellite or model data sets. However, since very good agreement over northern Africa is found for most of the indices, we can very probably exclude systematic measurement biases (like e.g. effects from the high surface albedo over the Sahara). Thus we conclude that the observed differences probably indicate deficiencies in the model simulations, possibly related to the sparseness of observational data over northern Africa used in the model. It is interesting to note that the differences are found for both oceanic and atmospheric indices which have rather different frequencies. These comparison results might help to improve the model performance over norther Africa (and to a lesser degree also over other regions).'

New Fig. 8:

[Figure]

Fig. 8: Fit coefficients for selected teleconnection indices, for which different patterns were found in the TCWV data set from satellite observations (left) and model simulations (right). The red circles indicate regions with substantial differences

More details about the changes are given in the individual replies to the Reviewer comments below:

Reviewer comment:
The manuscript selects extensive existing teleconnection indices and aims to identify teleconnection patterns in a new global dataset. It presents a new method to examine the reproducibility of the teleconnection in the global dataset along with other data sources. Although the manuscript first employs the dataset to the teleconnection research, which shows its novelty, the quality of the presentation needs substantial improvements. In the current version, the manuscript intends to address the research questions mentioned in the introduction, but the presentation of the results is confusing and difficult to follow. For example, the manuscript (e.g., Page 2, line 66-67) frequently mentioned the aim is to investigate the influence of teleconnections on the global distribution of the total column water vapor (TCWV). As far as I understand, the paper does not clearly address this issue. Could the authors clarify and stress the influence of teleconnection on the TCMV in the manuscript?

Author reply:
As mentioned in point B) above, we tried to make our aims more clear. Our aim was not to investigate the influence of teleconnection on the TCMV or to predict monthly TCWV using teleconnections. As described in point B) above, this was made more clear in many parts of the manuscript. For the text mentioned above (Page 2, line 66-67), we modified it to:
'In this study we investigate to which extent the temporal patterns of various teleconnections can be identified in the global distribution of the total column water vapor (TCWV).'

Reviewer comment:
Hence, I suggest some restructure of the body text of the manuscript. For instance, it might be beneficial to clarify the relationship between different groups of indices and their corresponding results.

Author reply:
We agree and made the corresponding changes, see points B), C), and D) above.

Reviewer comment:
Also, I suggest the authors improve the quality of the figures.

Author reply:
We agree and made the corresponding changes, see point D) above.

**Specific comments:**

Reviewer comment:
1. The manuscript used the water vapor column data from satellite observations in the red spectral range. Is there anything special for the use of red spectral range in the paper? It will be good to give some explanations otherwise I suggest removing "in the red spectral range" from the title.

Author reply:

We changed 'red' to 'visible'. The important point here is that the satellite observations observe scattered and reflected sun light. Thus they are sensitive for the total atmospheric column.

We added the following explanation to section 2.1:
' The data analysis is performed in the red spectral range. Since these satellite instruments observe scattered and reflected sun light, the observations are sensitive for the whole atmospheric column including the surface-near layers which usually contain the largest fraction of the total atmospheric TCWV.'

Reviewer comment:
2. Page 3, line 105: Could the authors provide one or two references, which shows that the variations of the TCWV are strongly associated with ENSO events? Or can authors provide the correlations over the tropical band?

Author reply:
The following references were added:
Simpson, J. J., J. S. Berg, C. J. Koblinsky, G. L. Hufford, and B. Beckley, The NVAP global water vapor data set: Independent crosscomparison and multiyear variability, Remote Sens. Environ., 76, 112–129, 2001.
Soden, B. J., The sensitivity of the hydrological cycle to ENSO, J. Clim., 13, 538– 549, 2000.
Wagner, T., S. Beirle, M. Grzegorski, S. Sanghavi, U. Platt, El-Niño induced anomalies in global data sets of water vapour and cloud cover derived from GOME on ERS-2, J. Geophys. Res, 110, D15104, doi:10.1029/2005JD005972, 2005.

Reviewer comment:
3. In the third section (Page 4, line 130-145), the authors did a great piece of work on putting various existing teleconnection indices together. The manuscript divides those indices into groups but indices in the same group can have high correlations, like ENSO indices (Fig. A3). The authors could focus on some selected indices and omit other highly correlated indices unless the differences among those indices affect the conclusion of the manuscript. It would be good to see more discussions in the line 143-144 for Fig.4.

Author reply:
In this comment we see two important aspects:
a) the description of the correlations and the grouping of the indices should be improved.
We followed this suggestion, see point C) above. In particular we added new sub groups of indices to table 2.
Furthermore, we added the information that for indices with high correlation similar spatial patterns are found in the TCWV data set. In section 4.1 we added the following information:
' As expected, for groups of indices with strong temporal correlation also similar spatial patterns are found. This is most obvious for indices similar to the ENSO index (first group of indices in Figures A6 and A8). Similar spatial

patterns are also found for other pairs of indices, e.g. between the Hawaiian Index (HAW) and the Pacific Decadal Oscillation (PDO) as well as between the South Tropical Atlantic index (STA) and the Equatorial Atlantic Index (EA_errst)'

b) It is suggested to ‚focus on some selected indices and omit other highly correlated indices unless the differences among those indices affect the conclusion of the manuscript.'

In principle we agree to this suggestion. However, in our opinion this was already addressed in a systematic way in the original manuscript by applying the orthogonalsiation of the teleconnection indices. Our choice to use an orthogonalsiation has the advantages that it is mathematically straight-forward and avoids any ambiguities and arbitrariness in the selection of the ‚best' index out of a group of similar indices. Overall our procedure should be seen as a two step approach: in the first step all available indices are used, because it is beforehand unclear, which of them are most significantly detected in the TCWV data set. But by applying our method to all indices, we can answer the question which indices are most significantly detected.

In a further step we then apply the orthogonalisation to obtain a new set of indices without any correlation amongst them.

To make our aims and the procedure more clear, we added the following information to the section 3:

' Many of these indices (describing the same phenomenon), but also many of the other teleconnection indices are highly correlated. The strength of these correlations is presented in Fig. 3 as a matrix with correlation coefficients between the different indices (after the seasonal cycles were removed). In spite of the correlations amongst the teleconnection indices, we decided as a first step to include them all in our study, because beforehand it is not clear which index might be best suited to represent a teleconnection phenomenon. Using our empirical approach, however, it becomes possible to quantify the significance and strength of the different indices and thus to select the best suited index for a given teleconnection phenomenon. Finally, we apply an orthogonalisation for the most significant indices (see section 7) to minimise the effect of the correlations and to identify the dominant temporal teleconnection patterns in our TCWV data set.'

In section 7 the explanation was extended to:

' To account for correlations between the different indices, we thus applied an orthogonalisation approach. For the orthogonalisation (based on the Gram–Schmidt process), all ‚significant' original indices and significant temporal derivatives (see Figure A11) were considered (in total 57 indices). The order of indices used in the iterative orthogonalisation process was from highest to lowest p99 values. The result of the orthogonalisation approach is a set of modified teleconnection indices, which shows zero correlation amongst each other (for the considered time period). Thus this new set of orthogonalised indices can be used to determine the number of independent significant teleconnection patterns in the global water vapor data sets.'

and in section 8:

'The cumulative delta RMS map for the orthogonalised indices represents the overall contribution of teleconnections to the variability of the global TCWV distribution.'

Reviewer comment:
4. In the fifth section, the manuscript uses the reversed indices but a clear explanation on the reversed indices and its meaning is needed, e.g. what is the meaning of 'reversed'. The current presentation makes Fig. 7 hard to understand.

Author reply:
The description of the reversed indices was made more clear, see point A) above. In section 5.1 the following clarification was added:
'The basic idea of our new approach is to use non-geophysical indices for the estimation of the significance level. Non-geophysical indices are indices without any temporal correlation with the temporal variations of the investigated geophysical data sets. For that purpose we chose all temporally reversed indices (see Table 2 and Fig. A6), because they cover all relevant frequencies of the true teleconnections. In practice, the time axis is flipped, that means the first entry (July 1995) will be assigned to the last month (October 2015), and so on.'

Reviewer comment:
5. Page 7, line 289-292: the results here are interesting. Could the authors provide more physical or dynamical explanations behind these results?

Author reply:
We added the following information to section 6:
'For the TCWV data sets, surface temperature and pressure, as well as most of the zonal winds, the largest p99 values are found for indices similar to ENSO. For the TCWV data sets and surface temperature, this can be expected, because the ENSO phenomenon is driven by the surface temperature (over the tropical Pacific). Accordingly, also the TCWV data sets will be strongly affected, because the TCWV depends strongly on the temperature in the lowest atmospheric layers. The strong influence of the ENSO phenomenon (BEST index) on the zonal winds at most levels can probably be explained by the fact that large scale phenomena like ENSO can have a strong influence on the quasi-persistent zonal flow patterns in the tropics and sub-tropics. For the geopotential heights and meridional winds, the largest p99 values are found for the polar atmospheric indices (mostly AAO, but also SCA). For the geopotential heights this might be expected because the polar atmospheric indices are defined based on anomalies of the geopotential heights. Why also for the zonal winds, the largest p99 values are found for the polar atmospheric indices is, however, is not clear to us.'

We added also a comparison of the maximum p99 values to section 6 (we also added a new column to table 3). The respective text in section 6 is:

' Our new method for the determination of the significance level also allows a direct comparison of the strengths at which the different indices are detected in the different data sets. In Table 3 also the maximum p99 values of the delta RMS normalised by the corresponding significance threshold values are shown. The highest normalised p99 values are found for the geopotential heights (except the 50hPa level) and the surface pressure. This finding is consistent with the fact that these quantities are used in most teleconnection studies and many indices are even defined using these quantities. The lowest normalised p99 values are found for zonal winds, for which also the smallest numbers of significant indices are obtained. Intermediate values are found for the water vapor data sets.'

Minor comments:

Reviewer comment:
1. Page 3, line 116-117: Could the authors add more descriptions for Fig. 2?

Author reply:
The following information was added:
'Similar patterns are found in all three data sets indicating the good consistency amongst them. The highest values are found over the tropics, especially over the west Pacific. Lower values are found towards higher latitudes showing the strong dependence of the TCWV on temperature.'

Reviewer comment:
2. Page 3, line 125: I suggest removing Figs. A1 and A2.

Author reply:
Both figures were removed as suggested.

Reviewer comment:
3. Page 4, line 138: The manuscript used the word 'fit' here, but the introduction of the fit function is shown in the next section. Moving the description of the fit function here might improve the clarity. Section 3 and 4 should be organized in a logical flow.

Author reply:
We reorganised sections 3 and 4 accordingly.

Reviewer comment:
4. Page 5, line 174: Could the authors clarify the meaning of a larger or lower value of delta RMS (eq. 3)?

Author reply:
After equation 3 the following information is added:
'The delta RMS is a measure for the magnitude of the variance of a considered data set, which can be explained by the chosen teleconnection

pattern. If there is high similarity of the temporal variation of an index with the temporal variation of the considered data set, the delta RMS values for both fits is large. If there is no similarity, the corresponding delta RMS value is zero.'

Also the following information is added:
'It should be noted that instead of the delta RMS values, also the correlation coefficients between the considered data set and the fit function (eq. 1) might have been used since the spatial patterns of both quantities are very similar (see Fig. A8).'

New Fig. A8:

[Figure]

**Fig. A8: Delta RMS (left) and r² values (right) for the fit of the ENSO index to the TCWV derived from satellite observations.**

Reviewer comment:
5. Table 2 and Figure 5 captions.

Author reply:
Corrected, many thanks!

---

## Author Comment (AC2) · 13 Dec 2020

Response to Reviewer 2

We thank the reviewer for the constructive and helpful remarks suggestions. We followed most of these suggestions as described below. Before we give our detailed answers to the individual comments, we first summary the most important changes to the manuscript.

The following major changes were made to the revised version of the manuscript:

**A) The motivation and description of the empirical method was improved.**

We agree with both reviewers that the description of new method had errors and was complicated. It was also not well motivated.

In the revised version we shifted many technical details of the new method to the appendix (e.g. the description of our normalisation approach as new appendix A1, or the investigation of the effect of time shifts as new appendix A3).

We also added more details about the fit function to section 4, and more details about the calculation of the temporally reversed indices to section 5.1.

For the motivation of our new approach we added the following information to section 5 (see also new Fig. 6):

[revised manuscript text omitted]

**C) The relationships between different indices and the motivation for the orthogonalisation of the indices was made more clear:**
We added new columns in Table 2 (see below). We now show separate columns for indices similar to ENSO, polar atmospheric indices, MJO indices, as well as other oceanic and atmospheric indices.

| Indices similar to ENSO (7) | Other oceanic indices (16) | Atmospheric polar indices (8) | MJO indices (15) | Other atmospheric indices (8) | Others indices (7) |
|---|---|---|---|---|---|
| | | | | | |

| | | | | | |
|---|---|---|---|---|---|
| BEST | HAW | SCA | MJ1 | PNA | Solar indices: |
| N34 | PDO | AAO | MJ2 | SOI | RI |
| TPI | PMM | EAWR | MJN | NOI | MGII |
| ONI | N1 | NAO | VPM1 | EA | SWO |
| ENSO | TNI | EPNP | VPM2 | QBO | S107 |
| N4 | NTA | AO | VPMN | Q30 | AP |
| IND | TNA | PE | RMM1 | Q50 | |
| | WHWP | WP | RMM2 | Q70 | HUR |
| | IPO | | RMMN | | (hurricane |
| | CAR | | OOMI1 | | frequency) |
| | AMO | | OOMI2 | | |
| | DMI | | OOMIN | | SAOD |
| | AMM | | FMO1 | | (stratospheric |
| | STA | | FMO2 | | AOD) |
| | TSA | | FMON | | |
| | EA_ersst | | | | |

In section 3 we added the following explanation:
'Many of these indices (describing the same phenomenon), but also many of the other teleconnection indices are highly correlated. The strength of these correlations is presented in Fig. 3 as a matrix with correlation coefficients between the different indices (after the seasonal cycles were removed). In spite of the correlations amongst the teleconnection indices, we decided as a first step to include them all in our study, because beforehand it is not clear which index might be best suited to represent a teleconnection phenomenon. Using our empirical approach, however, it becomes possible to quantify the significance and strength of the different indices and thus to select the best suited index for a given teleconnection phenomenon. Finally, we apply an orthogonalisation for the most significant indices (see section 7) to minimise the effect of the correlations and to identify the dominant temporal teleconnection patterns in our TCWV data set.'

To better motivate the orthogonalisation, we modified the respective information in section 7 to:
'To account for correlations between the different indices, we thus applied an orthogonalisation approach. For the orthogonalisation (based on the Gram–Schmidt process), all 'significant' original indices and significant temporal derivatives (see Figure A11) were considered (in total 57 indices). The order of indices used in the iterative orthogonalisation process was from highest to lowest p99 values. The result of the orthogonalisation approach is a set of modified teleconnection indices, which shows zero correlation amongst each other (for the considered time period). Thus this new set of orthogonalised indices can be used to determine the number of independent significant teleconnection patterns in the global water vapor data sets. We applied our new method to the new set of orthogonalised indices to test which of the modified indices have p99 values above the significance threshold.'

**D) The logical flow of the paper and the appearance was improved.**

As mentioned above several technical parts were shifted to the appendix. The science questions were better motivated in the introduction, and the corresponding answers were added to the conclusions.

Several Figures were shifted/deleted/modified:

-Fig. 3 was shifted to the appendix

-Fig. 8 was shifted to the appendix

-the upper part of Fig. 9 was shifted to the appendix

-Figs A1 and A2 were deleted as suggested

-the quality of Fig. A4 was improved and the number of the sub-figures was largely reduced (by a factor of 3)

-the quality of Fig. A9 was improved

**E) We added a new sub-section (6.1) for the comparison of the spatial patterns of the measured and simulated TCWV.**

While for most teleconnection indices very good agreement of the spatial patterns is found between the measuremed and simulated TCWV, for some indices also substantial differences are detected. These differences can point to shortcomings in either the satellite or model data sets (or both) and might be helpful for corresponding improvements.

We added a new Fig. 8 (see below) and the following new text:

'For most of the teleconnection indices, very similar spatial patterns are found in the TCWV data sets obtained from satellite or ECMWF data (see Fig. A9). This confirms both the high quality of the satellite measurements and model simulations. However, for some indices, also substantial differences are found (see Fig. 8). The most obvious differences are found over northern Africa. In principle, they could be caused by errors of both the satellite or model data sets. However, since very good agreement over northern Africa is found for most of the indices, we can very probably exclude systematic measurement biases (like e.g. effects from the high surface albedo over the Sahara). Thus we conclude that the observed differences probably indicate deficiencies in the model simulations, possibly related to the sparseness of observational data over northern Africa used in the model. It is interesting to note that the differences are found for both oceanic and atmospheric indices which have rather different frequencies. These comparison results might help to improve the model performance over norther Africa (and to a lesser degree also over other regions).'

New Fig. 8:

[Figure]

|  | Satellite observations | Model results |
|---|---|---|
| IPO | | |
| PDO | | |
| HAW | | |
| PMM | | |
| EPNP | | |
| PNA | | |

**Fig. 8: Fit coefficients for selected teleconnection indices, for which different patterns were found in the TCWV data set from satellite observations (left) and model simulations (right). The red circles indicate regions with substantial differences between the results for both data sets.**

More details about the changes are given in the individual replies to the Reviewer comments below:

Recommendation:
General comments: The authors are attempting to show how total column water vapor (TCWV) can be used to reveal the presence of atmospheric teleconnections seen in other datasets. This method is certainly interesting and could be of value, at least in the context of demonstrating the utility of TCWV in revealing existing teleconnections. However, the presentation in this paper was extremely difficult to follow as the authors jumped from one analysis to another with no clear direction as to why. There were many different technical approaches employed within this study, and while these likely have value in the context of what the authors' research goals are, the reasons for using the methods they employ were not well established.

Author reply:
We are thankful for this feedback and agree that our paper was partly difficult to read. We applied major restructuring and added missing information, see points B and D) above.

Reviewer comment:
Further, the authors state early in the study that they are going to compare the results with similar results from pressure, temperature, etc. fields more traditionally utilized in teleconnection studies. I did not see these comparisons.

Author reply:
It seems that here was a misunderstanding. The comparison to other data sets was one of the main aims of our study. The comparison results are shown and dicussed in section 6. We added more explanations for the findings of the comparisons to this section:
'For the TCWV data sets, surface temperature and pressure, as well as most of the zonal winds, the largest p99 values are found for indices similar to ENSO. For the TCWV data sets and surface temperature, this can be expected, because the ENSO phenomenon is driven by the surface temperature (over the tropical Pacific). Accordingly, also the TCWV data sets will be strongly affected, because the TCWV depends strongly on the temperature in the lowest atmospheric layers. The strong influence of the ENSO phenomenon (BEST index) on the zonal winds at most levels can probably be explained by the fact that large scale phenomena like ENSO can have a strong influence on the quasi-persistent zonal flow patterns in the tropics and sub-tropics. For the geopotential heights and meridional winds, the largest p99 values are found for the polar atmospheric indices (mostly AAO, but also SCA). For the geopotential heights this might be expected because the polar atmospheric indices are defined based on anomalies of the geopotential heights. Why also for the zonal winds, the largest p99 values are found for the polar atmospheric indices is, however, is not clear to us.'

We added also a comparison of the maximum p99 values to section 6 (we also added a new column to table 3). The respective text in section 6 is:
'Our new method for the determination of the significance level also allows a direct comparison of the strengths at which the different indices are detected in the different data sets. In Table 3 also the maximum p99 values of the delta

RMS normalised by the corresponding significance threshold values are shown. The highest normalised p99 values are found for the geopotential heights (except the 50hPa level) and the surface pressure. This finding is consistent with the fact that these quantities are used in most teleconnection studies and many indices are even defined using these quantities. The lowest normalised p99 values are found for zonal winds, for which also the smallest numbers of significant indices are obtained. Intermediate values are found for the water vapor data sets.'

Reviewer comment:
In general, the authors focused too heavily on the significance of the relationship between their empirical estimates of the TCWV using the teleconnection index and the TCWV itself. It read more like a study attempting to predict monthly TCWV using teleconnections, not a study linking TCWV to teleconnections. Either the study should be reframed in that context or the authors need to do a better job of linking their results back to the teleconnections they are trying to predict.
Which teleconnections were predicted well? Which were predicted poorly? Why? Such discussion was absent from this study and seems directly relevant to the research objectives outlined therein.

Author reply:
We are sorry that we gave the wrong impression here. As mentioned in point B) above, our aim was not to investigate the influence of teleconnection on the TCMV or to predict monthly TCWV using teleconnections. As described in point B) above, this was made more clear in many parts of the manuscript. In the introduction we modified the respective sentence (Page 2, line 66-67) to:
'In this study we investigate to which extent the temporal patterns of various teleconnections can be identified in the global distribution of the total column water vapor (TCWV).'

Specific comments:

Reviewer comment:
Most of the work done in PCA-based teleconnection studies in pressure/geopotential height is confined to midlatitude and Arctic regions in the Northern Hemisphere owing to the barotropic conditions in the tropical latitudes. This should be better specified by the authors.

Author reply:
This information was added to the introduction.

Reviewer comment:
If multiple indices characterizing the same phenomena exist (e.g. MJO, ENSO), why include them all? How do you reconcile the differences in how those indices are characterizing their teleconnection and relate those differences back to your results? (Lines 135-137).

Author reply:
The same aspect was also mentioned by the other reviewer, and we tried to make our motivation and strategy more clear in the revised manuscript:
Overall our procedure should be seen as a two step approach: in the first step all available indices are used, because it is beforehand unclear, which of them are most significantly detected in the TCWV data set. But by applying our method to all indices, we can answer the question which indices are most signifcantly detected.
In a further step we then apply the orthogonalisation to obtain a new set of indices without any correlation amongst them.
To make our aims and the procedure more clear, we added the following information to the section 3:
'Many of these indices (describing the same phenomenon), but also many of the other teleconnection indices are highly correlated. The strength of these correlations is presented in Fig. 3 as a matrix with correlation coefficients between the different indices (after the seasonal cycles were removed). In spite of the correlations amongst the teleconnection indices, we decided as a first step to include them all in our study, because beforehand it is not clear which index might be best suited to represent a teleconnection phenomenon. Using our empirical approach, however, it becomes possible to quantify the significance and strength of the different indices and thus to select the best suited index for a given teleconnection phenomenon. Finally, we apply an orthogonalisation for the most significant indices (see section 7) to minimise the effect of the correlations and to identify the dominant temporal teleconnection patterns in our TCWV data set.'

In section 7 the explanation was extended to:
'To account for correlations between the different indices, we thus applied an orthogonalisation approach. For the orthogonalisation (based on the Gram–Schmidt process), all ‚significant' original indices and significant temporal derivatives (see Figure A11) were considered (in total 57 indices). The order of indices used in the iterative orthogonalisation process was from highest to lowest p99 values. The result of the orthogonalisation approach is a set of modified teleconnection indices, which shows zero correlation amongst each other (for the considered time period). Thus this new set of orthogonalised indices can be used to determine the number of independent significant teleconnection patterns in the global water vapor data sets.'

and in section 8:
‚The cumulative delta RMS map for the orthogonalised indices represents the overall contribution of teleconnections to the variability of the global TCWV distribution.'

Reviewer comment:
In the fit functions, how were the quantities $c$ and $b$ determined? Were they based on a fit with the satellite data, the ERA, etc.? Nothing is provided in the text in this regard.

Author reply:
We checked the explanation of the quantities used in the fit function and added some more explanation. The definition of the involved quantities should now be more clear.

Reviewer comment:
The authors discuss the use of "reversed datasets" in section 5.1. However, they provide no discussion of what was reversed. Was it just the teleconnection time series? Was it the TCWV time series? Were they reversed in time? Did you just reverse the index numbers directly, as is done frequently in pattern recognition and database type work? I don't see why, if the reverse was temporal, why the correlations didn't simply change sign but remain the same magnitude. The authors need to provide a lot more explanation on this aspect of their study as they do not really describe it in much detail. Why did you do this?

Author reply:
Obviously our explanation of the details was not sufficient here. We added more explanations here, see also point A) above.
And in section 5.1 the following clarification was added:
'...Information about the significance of the fit results can be obtained from the fit function itself. However, in practice, the significance information from the fit has several limitations:
a) The determination of the significance is based on several assumptions about the data sets, e.g. that all data points of the time series have the same uncertainties and follow a normal distribution. However, the errors of the individual data points can be very different, e.g. the effect of clouds on the errors of the satellite TCWV data set can be very different for different seasons and regions. Also, the uncertainties are not only random but contain also systematic contributions. It is difficult (if not impossible) to quantify the uncertainties of the involved time series.
b) The determination of the significance is based on prescribed significance levels. The choice of such a significance level is arbitrary and the obtained significance information depends on this choice.
c) In several tests we fitted artificial time series to the TCWV data set. These tests showed that even for such non-geophysical time series 'significant' fit results can be obtained (see the examples in Fig. 6). On the left side of this figure, fit results for a time series containing only white noise, and on the right side fit results for a temporally reversed teleconnection index are shown (the temporally reversed index is obtained from the original index by mirroring the time axis). The blue and red areas show fit coefficients for both time series, which are classified as significant by the fit.
Based on these findings, we conclude the use of the significance of the detection of an index derived from the fit itself is not straight-forward.
To address these difficulties, we developed and applied an empirical approach to determine threshold values for the delta RMS values to decide whether an index is significantly detected in a global data set. The new procedure is described in the next section. It has the following two main advantages:

-the threshold values are determined empirically. Thus no assumptions on the properties of the time series or the significance levels have to be made.
- the method provides a clear procedure and in particular a metric which can be applied in a consistent way to different data sets and thus allows a quantitative comparison (see section 6).'

We also added information how the delta RMS values compare to the r² values at the end of section 5.1 (see also new Fig. A8):
'It should be noted that instead of the delta RMS values, also the correlation coefficients between the considered data set and the fit function (eq. 1) might have been used since the spatial patterns of both quantities are very similar (see Fig. A8).'

New Fig. A8:

[Figure]

**Fig. A8: Delta RMS (left) and r² values (right) for the fit of the ENSO index to the TCWV derived from satellite observations.**

Reviewer comment:
In section 8 the authors state they "orthogonalized" their indices. What method was used to do this? Why did they do this?

Author reply:
To better explain why we applied the orthogonalisation, we modifed and extended the information given in section 7. We also added the information of the orthogonalisation technique:
'To account for correlations between the different indices, we thus applied an orthogonalisation approach. For the orthogonalisation (based on the Gram–Schmidt process), all ‚significant' original indices and significant temporal derivatives (see Figure A11) were considered (in total 57 indices). The order of indices used in the iterative orthogonalisation process was from highest to lowest p99 values. The result of the orthogonalisation approach is a set of modified teleconnection indices, which shows zero correlation amongst each other (for the considered time period). Thus this new set of orthogonalised

indices can be used to determine the number of independent significant teleconnection patterns in the global water vapor data sets. We applied our new method to the new set of orthogonalised indices to test which of the modified indices have p99 values above the significance threshold.'

Technical corrections:

Reviewer comment:
The e.g. on line 51 can be removed.

Author reply:
Deleted

Reviewer comment:
What is a "time series like index"? (Line 78)

Author reply:
We replaced ,like' by ,such as' to make the meaning more clear.

Reviewer comment:
In Figures A1 and A2, are the times over which these averages were computed the same 1995-2015 time period? The ERA have a longer period of record so it would be good to specify this.

Author reply:
Both figures were deleted as suggested by the other reviewer.

Reviewer comment:
It is not clear why Figure A3 is included in the text. There are too many time series and their individual value in the study is not clear.

Author reply:
This figure was included for two reasons:
a) to add information about the sources of the different indices
b) to show the temporal patterns for the considered time period. This information is interesting for two reasons. First, the ,frequency' of an index can be directly recognised. Second, similarities in the temporal patterns can be easily seen.
For these reasons we decided to keep this figure in the manuscript.

Reviewer comment:
Figure A4 is almost impossible to read. There should be a compelling reason why this figure is included in the text as it includes well over 200 maps. The authors should choose which of those figures best illustrate their point and include those instead of including them all.

Author reply:

We agree that there are too many sub figures. And we want to apologise for the rather bad quality. In the revised manuscript we reduced the number of sub figures by a factor of 3 and improved the quality of the figure. We would like to keep this figure, because the global maps reveal many details of the spatial patterns found for the individual indices. It might be interesting for future studies to compare these patterns to similar results of their own analyses.

---

## Referee Report (RR1)

Second review of ACP-2020-565

Recommendation: Reject

General comments: While the authors did make strides to improve the explanation of the methods they employed, and their efforts did make their methods more clear, I obtained additional clarity in their approach and found several glaring concerns that need to be addressed. These concerns fall into three primary areas:

1) The blended use of teleconnections for specific spatial regions as a global product, even though many of these indices were not derived globally when they were created
2) The use of the ERA model data versus raw satellite data, and the subsequent use of their TCWV approach as effectively a verification measure for the ERA
3) Their empirical "reversed index" approach, which effectively is just a basic non-parametric hypothesis test.

The third of these issues impacted the interpretation of all results as it caused the authors to identify results as significant without proper context. Fixing this issue would require the authors to completely redo the analysis with more appropriate methods. As a result, I must still recommend rejection of this manuscript.

Major comments:

The authors did not really address my concern with the barotropic conditions in the tropics and why these regions are not used. In my experience deriving teleconnections, including barotropic regions generates large areas of high correlation due to the minimal height gradients in the tropics. Often these large areas wash out teleconnection features that would otherwise be present when using PCA (i.e. the first PC almost always exclusively identifies the tropics instead of a hemispheric teleconnection). As many teleconnections the authors considered (the NAO, PNA, etc.) were derived without including the tropics or the Southern Hemisphere, it is unclear how employing these teleconnections on a global study even makes sense, or even employing the teleconnections in the tropical latitudes which were not used in deriving the indices. How did you address this issue?

It seems like a strange methodological approach to verify the ERA model representations of TCVW using these teleconnection renderings (lines 87-90). Why not just directly verify the TCVW model data with the observation dataset? It seems like a very roundabout way to do model verification. Maybe a bigger question is why you are including the ERA data. Why not just use the satellite data directly? The differences do not seem that dramatic and there is no effort to explain why you did this except to compare model data against satellite, which does not help remove the "forecast" confusion in this study.

The use of the delta RMS quantity is strange. The authors even note (lines 209-210) that this quantity is basically the same as the correlation coefficient between the fit and the teleconnection. This makes sense as essentially you are doing a multivariate linear regression with an extra time term and you are just computing the variance explained by each teleconnection. This should scale almost exactly to just the correlation squared between the predicted value and the teleconnection. Why use this delta RMS instead of something simpler like $R^2$ to quantify the relationship between the teleconnection and the TCVW? Are there studies that employed a similar methodology?

I'm not sure your interpretation of the fit coefficient is correct. Are the time coefficients always the same for all teleconnections? If so, differences in magnitude in the teleconnection could be the reason for the change in the fit coefficient, not the actual amount of fit. You even show an example of this in Fig. A7, where the normalized ENSO index has a range from roughly +/-3 while the WHWP index has a range from +/-5. An almost 100% increase in magnitude in the index would affect the coefficient magnitude dramatically yet not explain any more variability.

With the volume of indices considered (Fig. A6), many of these have notable longer-term trends (MGII, Sunspots, PDO, AMO, etc.) where you do not even get an entire phase shift in your 20 year study period. How can you say with any certainty that there is a relationship without getting more than a single period of these quantities being in a "high" or "low" phase?

I would guess the small RMS values in the tropics are almost entirely a function of the barotropic conditions in the tropical regions (see my first comment above), not related to the impacts of clouds on the observations (as suggested in lines 222-224). Polar regions have similar issues as their conditions tend to be quasi-stationary. This alludes back to my earlier comment regarding the use of the tropics in this study.

While I technically agree with your statement on line 255, this is the nature of hypothesis testing. Most studies select a level (typically 95% or 99%) and go with it. I think the more important issue is trying to establish significance of these results using a hypothesis testing approach, since these tests are sensitive to sample sizes (e.g. you can get significance with a large sample size that could still have a poor relationship between the variables).

Are there citations of other studies that have used the "reversed index" approach you employed in this study? I have several concerns about it. First, reversing the time series does not ensure this is a completely uncorrelated relationship; random number generation would do a better job of that. As an example, I used a 70 year ONI time series and simply correlated the time series against its reverse and found a correlation of 0.16, which certainly is higher than what a random dataset should yield. This is even clear with several of your indices where the RMS values were fairly large considering this is supposed to emulate a "random" comparison. Second, the black dotted line is based on the mean and standard deviation of the reversed time series RMS 99[th] percentiles, yet when I look at the plot I see several points that would be "significantly" better than the reversed time series threshold. These points would also drive that mean upward as they are outliers (S107, MG11, etc.). This calls into question the validity of this approach since using a different statistics (e.g. the maximum) would cause almost all of your "significant" points to shift to non-significant. Maybe most importantly, this approach does not really show statistical significance. If you are treating this as a multiple-comparisons problem (which it appears you are), you need a Bonferroni correction on the cutoff threshold to ensure you are not committing type 1 errors (which are basically guaranteed with the 57 comparisons being done here). This would further shift the cutoff threshold upward and make more of your results non-significant. Why are you not using more traditional methods, such as bootstrapping, permutation testing, etc., to quantify this significance?

The latitudinal results in Fig. 11 make sense to me since most of the teleconnections related back to ENSO. Why are there so many teleconnections near the International Date Line? You never even discussed the longitudinal plot in the paper from what I could tell and that is a more interesting plot to me (and more difficult to explain).

The authors state that an advantage of their empirical approach is that it "avoids problems of existing algorithms for the determination of significance, because no assumptions on the significance level or the measurement uncertainties have to be made." However, by selecting the 99th percentile you have effectively created an $\alpha$ = 0.01 significance level as you are comparing your observation against an the 99th percentile of an empirical distribution. In effect you just did a hypothesis test, just with a slightly different appearance. If it is different than a hypothesis test it needs to be explained more effectively.

Minor comments:

Remove all of the uses of the word "like" in the e.g. statements (lines 46-47 and any others in the manuscript).

Line 98: "In section 2 the global datasets used in this study" is not a complete thought.

The y-axes in Fig. 7 should be consistent for both indices.

What do you mean by "reduced number" of data available? How small? What are the differences? (lines 228-229)

Would the result regarding zonal winds in the tropics not just be a consequence of the relationship between geopotential height and wind (lines 315-316)?

Why would a high surface albedo be a systematic measurement bias? Is the satellite instrument the one with the bias or the ERA data? (Lines 341-342).

There is strange comma use and formatting issues in section 7 of this paper.

Maybe I missed it, but why are there massive data gaps over Siberia and into India in the satellite data (Fig. 10)?

---

## Referee Report (RR2)

Review for acp-2020-565-manuscript-version2:

This manuscript has been greatly improved compared to its previous version. But there are still some minor issues in this paper.
Thus, I recommend acceptance of this manuscript after considering the following minor comments.

Line 17: obtained for ---- obtained from
Line 184: constant offset b and possible linear trend c ---- constant offset c and possible linear trend b

*For section 5 and 6: Considering the structure of the paper, I suggest that the author splits these sections into two parts and give a name for each part. For now, there has only 5.1 and 6.1. But it is just a personal writing style.*

---

## Author Response (AR2)

Dear Marc,

we uploaded our revised manuscript.
The most important change is that we applied (two) established methods for determining the significance of the different teleconnection indices (following the request of reviewer 1).
We also addressed all other comments of the reviewers. In cases where we disagree, or where we assume larger misunderstandings, we gave detailed explanations (see our detailed responses below).

Best regards,

Thomas
* * *
Reply to reviewer #1

Reviewer comment are shown in black
Our replies are shown in blue

First of all, we want to thank the reviewer for his/her second review.

In order to account for the request of using established methods for determining the significance of the different teleconnection indices, we performed two different tests: the so-called Walker test and the false discovery rate (FDR) test (see e.g. Wilks, 2006; 2016). The Walker test uses the minimum local p value as the global test statistic. The FDR test compares the p-values of all grid pixels with the distribution of the statistically expected FDR. Both tests deal with the question of "field significance", i.e. whether an index is significant anywhere. Another advantage of both tests is that they are rather robust with respect to spatial correlations of the input data. We also now used a generalized least squares fit accounting for temporal autocorrelation via a first-order autoregressive process. We found that these established statistical procedures (Walker and FDR test) yielded almost identical numbers of significant indices. The results were also very similar to those we derived from our empirical approach (for more details see below).
Thus, we hope to convince the reviewer (a) about the significance of our results and (b) the validity of our empirical approach, which is now complemented by established statistical methods in the revised version of our study.
We also addressed all other comments of the reviewer. In cases where we disagree, or where we assume larger misunderstandings, we gave detailed explanations.

Second review of ACP-2020-565

Recommendation: Reject

General comments: While the authors did make strides to improve the explanation of the methods they employed, and their efforts did make their methods more clear, I obtained additional clarity in their approach and found several glaring concerns that need to be addressed. These concerns fall into three primary areas:

1) The blended use of teleconnections for specific spatial regions as a global product, even though many of these indices were not derived globally when they were created

Several studies investigate the effect of teleconnections on a global scale, e.g. Hsu and Lin (1992) or Hoskins and Ambrizzi (1993). We also see no problem (instead we see it as an interesting application) to fit indices to regions far away from the regions for which they were originally defined. If an index is not important at a given location, the fit coefficient will be small. If, in contrast, the fit coefficient is found to be large, this is an interesting and potentially important hint that the teleconnection index might be of importance even far away from the region for which it was originally defined. Here it should be noted that for ENSO or QBO, influences were indeed found far away from the regions where they were defined.
We would see it as an unjustified pre-selection to restrict the analyses of the individual indices to limited regions. Even if no new discoveries are made, we see at least no negative consequences by performing the fits of all indices on a global scale.
We should maybe also clarify, that If we find an index to be significant, this does not mean that it is significant everywhere on the globe.

2) The use of the ERA model data versus raw satellite data, and the subsequent use of their TCWV approach as effectively a verification measure for the ERA

Here two aspects are important:
a) Why were the model data included at all in the study?
The satellite data are derived for the overpass times and for (mostly) clear sky conditions. Therefore, it was important to test whether the satellite data can be seen as representative for all times and conditions. To answer this question, model data are well suited, because results for specific selections can be compared, e.g. results for all data or only clear sky daytime conditions. We made such comparisons based on ERA-interim data and could conclude that the satellite measurements can be seen as representative for the all-sky conditions.
b) As discussed in Beirle et al., 2018, the long term satellite data set was created with a focus on temporal stability (and consistency between the different sensors). To achieve this aim, for the data analysis a simplified analysis procedure was applied (e.g. by using the oxygen absorption in the red spectral range instead of operational cloud products). It was shown in many studies that especially the standard cloud products of satellite instruments are often affected by instrument degradation and are also not consistent between the different sensors. These problems are overcome by our simplified analysis procedure. This analysis procedure, on the other hand, also has its drawbacks. Especially for individual measurements, the uncertainties can be rather large. Also systematic biases might occur for specific regions, e.g. related to the effect of the surface albedo. These limitations are not important for climate and trend studies, and by comparison with independent data sets, the temporal stability of our global long term satellite data set was found to be very good. However, a direct quantitative comparison to the model data would not make much sense, because of the known systematic biases of our satellite data set.
The main advantage of our satellite data set is that it is very well suited to investigate the temporal variability, e.g. related to teleconnections. Within the scope of this study, the same teleconnection analyses were performed for the satellite and model data. For almost all indices, astonishingly good consistency of the derived spatial patterns and the significance of the different teleconnection indices was found, indicating that the temporal patterns are well covered in both satellite and model data sets.

To make our motivation for the use of the model data in our study more clear, we added the following text in section 2.1 '.The main purpose of using model data is that we want to see if teleconnections are found in a similar way in both satellite and model data sets. In addition, the use of model data also allows to quantify a possible clear-sky bias in the satellite observations, because these observations are made for mainly cloud-free conditions. Therefore we use two data sets:'

3) Their empirical "reversed index" approach, which effectively is just a basic non-parametric hypothesis test.

As suggested by the reviewer, we now applied established methods for testing the significance of teleconnection indices to our global water vapor data set. For that purpose we chose the so-called Walker test and the false discovery rate (FDR) test (e.g. Wilks, 2006; 2016). We also accounted for temporal autocorrelation effects within the fit method by assuming an AR(1) process of the fit residual (Seabold and Perktold, 2010). From the fit results we derived the local p-values (by a two-tailed t-test) of each fitted index for every grid cell. To account for the effect of test multiplicity, we then applied the Walker and FDR test to the global distribution of p-values for each teleconnection index.

Both tests are well suited for global applications as they are robust to the effects of spatial correlation. Interestingly, especially for the FDR test very similar results were obtained compared to our empirical approach, see figure below. Only a few indices with low frequencies (Q50, Q70, and IPO) which were previously be found to be slightly above the significance level, are now found to be slightly below the significance level. Conversely, some previously non-significant indices with high frequencies are now found to be slightly above the significance threshold. These changes are related to the fact that for the new method we also used a modified fit explicitly considering the temporal correlations of the indices.

The most important finding is that these differences between the FDR test and the old method are only found for indices close to the significance thresholds and don't affect the main findings of the paper. The number of significant indices found for the old and new method differs only by 2 (42 for our empirical method and 44 for the new method) if one takes into account that the OOMI2 and FMO2 as well as the OOMI1 and FMO1 indices are very similar.

Because of the good agreement between our empirical approach and the new standard method, we decided to keep the results of our empirical method in the paper. We added a new section to the paper (section 5.2) which presents and discusses the comparison results between the empirical method and the well established statistical methods from literature.

[Figure]

New Fig. 8 Comparison of the results of the three approaches. Top: results for the Walker test (left axis) and FDR test (right axis). The indices are sorted according to the results of the Walker test. The black vertical line indicates the significance threshold (similar for both tests). Middle: comparison of the results for the Walker test (left axis) and our empirical (reverse) approach (right axis). The indices are sorted according to the results of the Walker test. The black vertical line indicates the significance threshold of the Walker test. The red horizontal line indicates the significance threshold for our empirical approach. Bottom: similar as for the middle panel, but for the FDR test.

The third of these issues impacted the interpretation of all results as it caused the authors to identify results as significant without proper context. Fixing this issue would require the

authors to completely redo the analysis with more appropriate methods. As a result, I must still recommend rejection of this manuscript.

For our main reply to this point, please see our previous comment above.
Concerning the interpretation of significance, we want to point out that 'significance' does not mean that an index is significant everywhere on the globe. Our interpretation of significance is that of 'field significance' as described e.g. by Wilks (2006). We added the following clarification in the introduction: 'Here it should be noted that with significance we don't mean that an index is significant everywhere on the globe. We are rather interested in whether an index is significant somewhere on the globe, the so-called 'field significance' (see e.g. Wilks, 2006).'

**Major comments:**

The authors did not really address my concern with the barotropic conditions in the tropics and why these regions are not used. In my experience deriving teleconnections, including barotropic regions generates large areas of high correlation due to the minimal height gradients in the tropics. Often these large areas wash out teleconnection features that would otherwise be present when using PCA (i.e. the first PC almost always exclusively identifies the tropics instead of a hemispheric teleconnection).

For us, this comment is not completely clear. But probably there is a misunderstanding here: our aim was not to derive teleconnections. We simply tested how strong existing teleconnection indices correspond to the variabilty of the water vapor distribution on a global scale. Such global studies are not unusual, see e.g. Hsu and Lin (1992) or Hoskins and Ambrizzi (1993).

As many teleconnections the authors considered (the NAO, PNA, etc.) were derived without including the tropics or the Southern Hemisphere, it is unclear how employing these teleconnections on a global study even makes sense, or even employing the teleconnections in the tropical latitudes which were not used in deriving the indices. How did you address this issue?

As mentioned in our reply to the first overall point, we see no problem (and instead only advantages) by analysing all indices for all grid points of the global data sets.
If one looks at the derived patterns in Fig. A9, one can clearly see that in the tropics especially for many atmospheric indices, indeed low fit coefficients are obtained. In Fig. A12, it can be seen that after the normalisation, the fit coefficients in the tropics are further strongly reduced. We find these results interesting and relevant. They can only be obtained if the indices are fitted to the data sets globally. In our opinion, there is really no ‚issue' here.

It seems like a strange methodological approach to verify the ERA model representations of TCVW using these teleconnection renderings (lines 87-90). Why not just directly verify the TCVW model data with the observation dataset?
It seems like a very roundabout way to do model verification. Maybe a bigger question is why you are including the ERA data. Why not just use the satellite data directly?

See also our reply to the first general point above. Our intention was not to verify the ERA model. The main aim was to see whether there is a systematic clear sky bias in the satellite observations. From the satellite to model comparison we can conclude that no such bias is found.

The differences do not seem that dramatic and there is no effort to explain why you did this except to compare model data against satellite, which does not help remove the "forecast" confusion in this study.

In our opinion, the "forecast" confusion of the first version of the manuscript should have been clarified in the first revised version. It was never our intention to make forecasts.

The use of the delta RMS quantity is strange. The authors even note (lines 209-210) that this quantity is basically the same as the correlation coefficient between the fit and the teleconnection. This makes sense as essentially you are doing a multivariate linear regression with an extra time term and you are just computing the variance explained by each teleconnection. This should scale almost exactly to just the correlation squared between the predicted value and the teleconnection. Why use this delta RMS instead of something simpler like R2 to quantify the relationship between the teleconnection and the TCVW? Are there studies that employed a similar methodology?

The reason to use the delta RMS is that the results for the individual indices directly quantify the contribution by that index to the variability of the data set. Thus, it quantifies the *relevance* of the different indices, (while significance alone does not imply that a signal is relevant). This is especially important when these contributions are summed up to derive the cumulative effect of all indices (e.g. for the orthogonalised indices).

We found no other studies which used the delta RMS value for the determination of significance. But quantities very similar to the delta RMS were used in several studies to quantify the individual or cumulative contributions of different temporal patterns to the total variance (e.g. Horel, 1981; Trenberth and Paolino, 1981; Barnston and Livezey, 1987). There, the aim was to determine the order of importance of the different time series. This aim is similar to our study.

We also want to note here that we applied our ,reversed index method' also to the r² values (instead of the delta RMS), and we found exactly the same number of ,significant' indices (only two indices with delta RMS values slightly below the threshold were now found to be slightly above the r² threshold (and vice versa).

I'm not sure your interpretation of the fit coefficient is correct. Are the time coefficients always the same for all teleconnections? If so, differences in magnitude in the teleconnection could be the reason for the change in the fit coefficient, not the actual amount of fit. You even show an example of this in Fig. A7, where the normalized ENSO index has a range from roughly +/-3 while the WHWP index has a range from +/-5. An almost 100% increase in magnitude in the index would affect the coefficient magnitude dramatically yet not explain any more variability.

Note that the TCWV anomaly was detrended and deseasonalized before the fit, such that the time coefficient is negligible.
In order to make the indices comparable, we have normalized them to their respective standard deviation, which is a common procedure (see e.g. Horel, 1981). The absolute fit coefficient is not of main focus in our study, as the importance of a teleconnection is quantified by the delta RMS, which is not affected by a scaling of the index.

With the volume of indices considered (Fig. A6), many of these have notable longer-term trends (MGII, Sunspots, PDO, AMO, etc.) where you do not even get an entire phase shift in your 20 year study period. How can you say with any certainty that there is a relationship without getting more than a single period of these quantities being in a "high" or "low" phase?

It is true that for some of the considered indices no complete phase is covered by the 20 year study period. Nevertheless, we don't see this as a reason not to include these indices. Even if not a full period is covered by the 20 year period, it is still interesting to investigate whether the temporal signature is found in the considered global data set. Of course, it is well possible that potential teleconnections caused by effects with very low frequency could not be detected by the 20 year record we are using. We have added a the following text to section 3: 'Here it should be noted that for some of these indices with low frequencies (e.g. MGII or IPO) no full period is covered by our 20 year-long satellite TCWV data set, which might be a reason why they are not significantly detected'.

I would guess the small RMS values in the tropics are almost entirely a function of the barotropic conditions in the tropical regions (see my first comment above), not related to the impacts of clouds on the observations (as suggested in lines 222-224). Polar regions have similar issues as their conditions tend to be quasi-stationary. This alludes back to my earlier comment regarding the use of the tropics in this study.

We think there is a misunderstanding here. In the paper we wrote with respect to the effect of clouds: 'In mid-latitudes, systematically higher RMS are found for the satellite observations compared to the model results. This is probably related to the rather large effects of clouds on the satellite observations, which becomes especially important in these regions (clouds lead to less valid observations and larger measurement uncertainties).'.
We did not say that the RMS is in general high in mid-latitudes because of clouds, but that is it higher there for the satellite data set compared to the model data.
We think our description and explanation of the cloud effects on the satellite data set was reasonable and correct. We see no need to change it.

While I technically agree with your statement on line 255, this is the nature of hypothesis testing. Most studies select a level (typically 95% or 99%) and go with it. I think the more important issue is trying to establish significance of these results using a hypothesis testing approach, since these tests are sensitive to sample sizes (e.g. you can get significance with a large sample size that could still have a poor relationship between the variables).

We followed the suggestion to apply standard hypothesis testing approaches, see our comments to the first points above.

Are there citations of other studies that have used the "reversed index" approach you employed in this study?

To our knowledge, we are the first who used this approach.

I have several concerns about it. First, reversing the time series does not ensure this is a completely uncorrelated relationship; random number generation would do a better job of that.

We fully agree with the statement that ,reversing the time series does not ensure this is a completely uncorrelated relationship'. We obtained exactly the same finding in our study (see Fig. A2).
In order to address the effect of such correlations, we applied the method described in appendix 2: all indices with high correlations to important original indices are removed before the delta RMS threshold is determined. This removal is crucial. If these indices were not removed, the resulting thresholds would indeed be much too high.

As suggested by the reviewer, we also used a large number (100) of random time series to determine the significance threshold. The derived threshold value was slightly smaller (0.0027) than that from the reversed time series (0.0031). This indicates that the threshold from the reversed time series is the more conservative estimate.
We modified the already existing information about the threshold derived from (only few) random time series by the following text: 'We also applied the same method to a set of 100 artificial random time series and obtained a slightly smaller threshold value of 0.0027 indicating that the threshold value obtained from the temporally reversed time series is reasonable.'

As an example, I used a 70 year ONI time series and simply correlated the time series against its reverse and found a correlation of 0.16, which certainly is higher than what a random dataset should yield. This is even clear with several of your indices where the RMS values were fairly large considering this is supposed to emulate a "random" comparison.

We found the similar behavior for several indices, see Fig. A2. Therefore we applied the method described in appendix 2). As mentioned above, this procedure is essential for the application of the reversed index approach. One important advantage of the reversed index approach is that it covers all frequencies of teleconnections.

Second, the black dotted line is based on the mean and standard deviation of the reversed time series RMS 99th percentiles, yet when I look at the plot I see several points that would be "significantly" better than the reversed time series threshold. These points would also drive that mean upward as they are outliers (S107, MG11, etc.). This calls into question the validity of this approach since using a different statistics (e.g. the maximum) would cause almost all of your "significant" points to shift to non-significant.

The high values mentioned by the reviewer belong to exactly the indices with high correlations to important original indices. As mentioned above these indices were removed before the calculation of the thresholds (see appendix 2). This step is crucial for the application of the reversed index approach.

Maybe most importantly, this approach does not really show statistical significance. If you are treating this as a multiple-comparisons problem (which it appears you are), you need a Bonferroni correction on the cutoff threshold to ensure you are not committing type 1 errors (which are basically guaranteed with the 57 comparisons being done here). This would further shift the cutoff threshold upward and make more of your results non-significant. Why are you not using more traditional methods, such as bootstrapping, permutation testing, etc., to quantify this significance?

As stated above, the main aim of the study is not only to strictly determine whether an index is significantly detected or not, but to establish an order of importance. We find that this aim

is well achieved by our method. This is also confirmed by the good agreement with the results of the standard hypothesis testing approaches, see above.

The latitudinal results in Fig. 11 make sense to me since most of the teleconnections related back to ENSO. Why are there so many teleconnections near the International Date Line? You never even discussed the longitudinal plot in the paper from what I could tell and that is a more interesting plot to me (and more difficult to explain).

Many thanks for this hint! This is indeed an interesting finding, to which we had a closer look. The location of 4 of the orthogonalised indices found for longitudes between –167° and –180° is located at latitudes between 38 and 71°N.
We searched for similar findings in other studies and found that in several publications, enhanced activity was found in the same area (e.g. Hsu and Lin, 1992; Hoskins and Ambrizzi, 1993; Trenberth et al., 1998). One possible reason for the enhanced activity in this area might be the effect of jet exit regions, which is driven to a large extent by the Earth's topography (Feldstein and Franzke, 2017). We added this information to section 8.

The authors state that an advantage of their empirical approach is that it "avoids problems of existing algorithms for the determination of significance, because no assumptions on the significance level or the measurement uncertainties have to be made." However, by selecting the 99th percentile you have effectively created an $\alpha = 0.01$ significance level as you are comparing your observation against an the 99th percentile of an empirical distribution. In effect you just did a hypothesis test, just with a slightly different appearance. If it is different than a hypothesis test it needs to be explained more effectively.

We tested whether our results depend on the exact choice of the percentiles. For that purpose we repeated our approach using the 95th and 98th percentiles of the delta RMS and compared the results to the original results (for the 99th percentiles). For all three cases, exactly the same number of 'significant' indices (40) was found, clearly indicating that our results do not critically depend on the exact choice of the percentile. This can be understood by the fact that for both the original and the reversed indices the same percentiles are used. Thus the effect of different percentiles almost cancels out. This indicates the robustness of our empirical method.
We added the following text in section 5.1: 'Here it should be noted that the exact choice of the percentile is not critical, as the same percentile is applied to both original and reversed indices. We found exactly the same set of significant indices (see below) if we used the 95th percentile or the 98th percentile.'

**Minor comments:**

Remove all of the uses of the word "like" in the e.g. statements (lines 46-47 and any others in the manuscript).

The 'like' in the 'e.g. statements' were deleted.

Line 98: "In section 2 the global datasets used in this study" is not a complete thought.

corrected

The y-axes in Fig. 7 should be consistent for both indices.

This statement is not completely clear to us, because the same y axis is used for the original and reversed indices. Maybe the reviewer wanted to suggest to use the same y-axes for the three data sets shown in Fig. 3. We changed the y-axis of the middle panel ('ECMWF all data') to be consistent with the bottom panel. We also added a hint in the figure caption that different y axes are used for the satellite and model data.

What do you mean by "reduced number" of data available? How small? What are the differences? (lines 228-229)

We modified the sentence to: 'The RMS for the model results for clear sky conditions is slightly higher than for the model results for all conditions, which is to be expected because of the reduced number of input data for the cloud-filtered data set (about 40% less compared to the non-filtered data set).'
This information is also added to section 2.1.

Would the result regarding zonal winds in the tropics not just be a consequence of the relationship between geopotential height and wind (lines 315-316)?

We added this information to the text:
'Also for the zonal winds, the largest p99 values are found for the polar atmospheric indices, which is probably caused by the strong relationship between geopotential heights and winds.'

Why would a high surface albedo be a systematic measurement bias? Is the satellite instrument the one with the bias or the ERA data? (Lines 341-342).

To make this more clear, we changed 'biases' to 'biases in the satellite data set'.

We added the following information to section 2.1: 'It should be noted that the satellite data set used in this study was optimized with respect to temporal stability, which makes it well-suited for climate studies. However, because of the rather simple analysis approach, for specific situations small systematic biases of the absolute values might occur, e.g. related to the effects of surface albedo or terrain height.'

There is strange comma use and formatting issues in section 7 of this paper.

Here it is not clear, which strange use of commas is meant by the reviewer. However, we want to point out that the use of English language will be checked by the editorial office before final publication.
We removed the gap between the last sentence and the main text in section 7.

Maybe I missed it, but why are there massive data gaps over Siberia and into India in the satellite data (Fig. 10)?

Part of the gaps are caused by data loss due to calibration of the satellite instrument which takes place north of India. Additional gaps are caused above high mountains by the cloud filter applied to the satellite instruments.
We added this information to section 2.1.

Reply to reviewer #2

First of all, we want to thank the reviewer for his/her second review.

Review for acp-2020-565-manuscript-version2:

This manuscript has been greatly improved compared to its previous version. But there are still some minor issues in this paper.
Thus, I recommend acceptance of this manuscript after considering the following minor comments.

We thank the reviewer for the positive assessment.

Line 17: obtained for ---- obtained from

Corrected

Line 184: constant offset b and possible linear trend c ---- constant offset c and possible linear trend b

Corrected

For section 5 and 6: Considering the structure of the paper, I suggest that the author splits these sections into two parts and give a name for each part. For now, there has only 5.1 and 6.1. But it is just a personal writing style.

For section 5, the issue was ‚resolved' by adding the new section 5.2.

For section 6, we added a new subsection for the first part:
‚6.1 Comparison of the results for the TCWV data sets to those for the other data sets'

References

Hoskins, B. J., & Ambrizzi, T. (1993). Rossby Wave Propagation on a Realistic Longitudinally Varying Flow, Journal of Atmospheric Sciences, 50(12), 1661-1671.

Hsu, H., & Lin, S. (1992). Global Teleconnections in the 250-mb Streamfunction Field during the Northern Hemisphere Winter, Monthly Weather Review, 120(7), 1169-1190.

Seabold, Skipper and Josef Perktold, Statsmodels: Econometric and Statistical Modeling with Python, Proceeding of the 9th Python in science conference (SCIPY 2010) http://conference.scipy.org/proceedings/scipy2010/pdfs/seabold.pdf, 2010.

Trenberth, K.E., G.W. Branstator, D. Karoly, A. Kumar, N.-C. Lau, C. Ropelewski, Progress during TOGA in understanding and modeling global teleconnections associated with tropical sea surface temperatures, J. Geophys. Res., 103, 14291-14324, 1998.

Wilks, D. S. (2006). On "Field Significance" and the False Discovery Rate, Journal of Applied Meteorology and Climatology, 45(9), 1181-1189.

Wilks, D. S. (2016). "The Stippling Shows Statistically Significant Grid Points": How Research Results are Routinely Overstated and Overinterpreted, and What to Do about It, Bulletin of the American Meteorological Society, 97(12), 2263-2273.